# Transglutaminase 2-expressing macrophages modulate adipose tissue inflammation
Diana M. Elizondo[1,3], Tushar P. Patel [1], Benjamin T. Cole [1], Eliza A. Jansujwicz[1], Jocelyn Chen[1], Melanie C. Hollis[1], Apratim Mitra[2] & Jack A. Yanovski [1] ✉

We investigated Transglutaminase 2 (TGM2) in high fat diet (HFD) obese mice, finding upregulated TGM2+ adipose tissue macrophages (ATMs) in HFD epididymal white adipose tissue (eWAT) compared to chow diet (CD) eWAT. Using *Tgm2* CRISPR silencing, we examined TGM2 modulation of inflammation in vitro within bone marrow-derived macrophages (BMMs), as well as in co-cultured eWAT stromal vascular fraction (SVF) cells. Tgm2 silencing in BMMs led to increased pro-inflammation, compared to control. In contrast, in vitro exposure of eWAT SVF to recombinant TGM2 increased anti-inflammatory IL-10 secretion. However, IL-10 was not induced by recombinant TGM2 in CD activated CD4 + T cells, or in HFD-derived SVF CD4 + T cells. In vivo *Tgm2* silencing in CD11b+ cells in HFD mice resulted in pro-inflammation in eWAT and serum, and increased adiposity and insulin resistance, suggesting that TGM2 + ATMs possess an anti-inflammatory role in obesity that is insufficient to reverse obesity-induced inflammation.

Low-grade chronic inflammation has been mechanistically linked to insulin resistance (IR) and metabolic dysfunction in individuals with obesity[1–3]. It is characterized by an increase in pro-inflammatory cytokines, and the infiltration of adipose tissue (AT) with pro-inflammatory immune cells, such as macrophages.

AT macrophages (ATMs) are polarized into pro- or anti-inflammatory states, often referred to as M1 or M2, respectively[4]. Classically, during the lean state, F4/80 + CD11c-CD206 + CD64 + ATMs possess an anti-inflammatory profile[5]. Their main function is to create a homeostatic environment via anti-inflammatory cytokines, such as IL-10, a cytokine demonstrated to regulate obesity-associated metabolic dysfunction[6]. Collectively, this helps to resolve inflammation, preserve normal adipocyte metabolism, and clear apoptotic adipocytes via efferocytosis. In obesity, a high rate of adipocyte stress and death during AT remodeling events leads to a pro-inflammatory polarization shift. The pro-inflammatory ATMs are typically differentiated from the anti-inflammatory subset based on CD11c marker expression[7,8], and have been previously reported as key contributors to IR in obesity[9]. However, literature reports now suggest that ATMs in obesity acquire a malleable (or plastic-like) phenotype with the development of populations possessing a mixture of M1 and M2 polarization markers in response to AT environmental cues, particularly during tissue remodeling

events (i.e., rapid tissue expansion)[10–12]. Therefore, the classic M1/M2 macrophage model has been challenged by new evidence showing multiple ATM subsets in obesity[7,13], whose origins and functions remain to be fully established.

Transglutaminase 2 (TGM2) is a protein ubiquitously expressed in multiple cell types, including monocytes, M2 macrophages[14–19], thymocytes[20], myoblasts[21], endothelial cells[22], and preadipocytes[23] among others. It has been shown to play dual roles in supporting anti-inflammatory macrophage properties, such as efferocytosis functions[24–26] and modulating LPS pro-inflammatory responses[27]. This multifunctional protein has both enzymatic and scaffolding properties that control inflammatory responses, cell differentiation and activation, blood coagulation, skin barrier formation, and extracellular matrix assembly. Accordingly, TGM2 has been associated with the pathophysiology of autoimmune disorders and tissue degenerative diseases[14]. Studies have shown that TGM2 can be induced by hyperglycemia[28], and can modulate IR and adipogenesis[23,29]. Collectively, these findings suggest a potential role for TGM2 in regulating AT homeostasis. However, the role of macrophage-derived TGM2 paracrine effects on AT inflammation remains to be fully elucidated.

We investigated the role of TGM2 in modulating macrophage inflammation, along with the paracrine effects of secreted macrophage-

[1]Section on Growth and Obesity, Division of Intramural Research, Eunice Kennedy Shriver National Institute of Child Health and Human Development (NICHD), National Institutes of Health, Bethesda, MD, USA. [2]Bioinformatics and Scientific Programming Core, Eunice Kennedy Shriver National Institute of Child Health and Human Development (NICHD), National Institutes of Health, Bethesda, MD, USA. [3]Present address: Department of Biological Sciences, University of Maryland Baltimore County, Baltimore, MD, USA. ✉e-mail: jy15i@nih.gov

derived TGM2 on AT immune cells' inflammatory profile. We herein report increased TGM2 expressing ATMs in eWAT from high-fat diet (HFD)-induced obese male mice and showed that silencing of *Tgm2* in bone marrow-derived macrophages (BMMs) resulted in depressed anti-inflammatory profiles within *Tgm2* CRISPR silenced BMMs and in co-cultured AT stromal vascular fraction (SVF) leukocytes. In turn, ex vivo administration of TGM2 recombinant protein to SVF induced IL-10 anti-inflammatory expression and release in homeostatic T cells. Lastly, an in vivo mouse model silencing *Tgm2* in CD11b myeloid cells in HFD mice further demonstrated that reducing TGM2 in myeloid cells results in increased pro-inflammatory responses, along with increased obesity and IR.

Taken together, our results demonstrate that TGM2 expression in macrophages exerts anti-inflammatory properties in AT lymphocytes, thereby protecting against the development of IR. These data highlight the role of TGM2-producing macrophages in balancing the AT inflammation and metabolic dysfunction induced by HFD obesity.

## Materials and methods
A complete list of materials and reagents used is given in Supplemental Table 1.

### Animals
C57BL/6 J mice were purchased from The Jackson Laboratory (Strain Number 000664), housed in pathogen-free facilities and fed ad libitum either a chow diet (CD: LabDiet PicoLab, 5L0D; 4.09 kcal/gm, 29.8% protein, 13.4%fat, 56.7% carbohydrate) or a HFD (60% HFD: Research Diets, D12492; 5.24 kcal/gm, 20% protein, 60% fat, 20% carbohydrate) beginning at 6 weeks of age for a total of 10 weeks (until 16 weeks of age). For in vivo *Tgm2* silencing studies, B6.129S1-*Tgm2*[tm1Rmgr]/J mice were purchased from The Jackson lab (Strain Number 024694), housed in pathogen-free facilities and fed ad libitum (CD: LabDiet PicoLab, 5L0D; 4.09 kcal/gm, 29.8% protein, 13.4%fat, 56.7% carbohydrate) until 6 weeks of age and were given an HFD (60% HFD: Research Diets, D12492; 5.24 kcal/gm, 20% protein, 60% fat, 20% carbohydrate) for a total of 5 weeks (until 12 weeks of age).

### Sex as a biological variable
C57BL/6 J or B6.129S1-*Tgm2*[tm1Rmgr]/J male mice were utilized due to augmented pro-inflammatory responses to obesity compared to females, making diet-induced obese males a better model for these studies[30–32]. The B6.129S1-*Tgm2*[tm1Rmgr]/J mouse cohort used for in vivo silencing of *Tgm2* in myeloid cells was further complemented with a female cohort to assess potential sexual dimorphisms in observed phenotype.

### In vivo *Tgm2* silencing in CD11b myeloid cells
Lentivirus carrying CD11b upstream of Cre-recombinase with EGFP reporter downstream of CMV promoter pLV[Exp]-EGFP-CD11b>Cre [Cat:VB230425-1457ejv] (pLV.CD11b-Cre), or respective lentivirus control carrying human EF1A promoter upstream of a mCherry reporter to detect potential non-specific gene induction, followed by CMV promoter upstream of EGFP reporter pLV[Exp]-EGFP/Puro-EF1A>mCherry [Vector ID: VB010000-9298rtf] (pLV.Control) were purchased from Vector Builder (Chicago, IL). EGFP was utilized to evaluate transduction efficiency. Purchased lentiviruses were ultra-purified for in vivo applications, diluted in sterile PBS to a final concentration of $10^8$ TU/mL in 250 uL. Lentiviruses were injected intraperitoneally in 5 week old *Tgm2* floxed homozygous mice that were randomized by chance. Mice were placed on a 60%HFD 1-week post-pLV injections and monitored for changes in body weight, glucose and insulin responses by glucose or insulin tolerance (ITT) tests, along with changes in whole body composition by DEXA scan.

### Metabolic evaluations
Total mouse and adipose tissue (AT) depot weights were measured using a standard balance. Glucose homeostasis was evaluated via tail vein prick at the end of diet-treatment with a glucometer (Glucometer Elite, Bayer, Elkhart, IN) under non-fasting conditions. Markers of inflammation were measured by qPCR and Luminex assay. Glucose (post-16h fasting conditions) and Insulin (post-4h fasting conditions) Tolerance tests were performed by intraperitoneally injecting 2.5 g/kg body weight glucose, or 0.75 U/kg body weight insulin, respectively into CD or HFD mice. Glucose concentrations were measured every 30 min for 120 min by glucometer. Digital *X*-ray system Ultra focus DXA Faxitron (Tucson, AZ) was used for evaluation of whole-body composition in mice post-lentivirus and diet-treatments.

### Real time quantitative reverse transcription PCR
Total epididymal white adipose tissue (eWAT), SVF, primary adipocytes, or cultured cells were homogenized by bead homogenizer lysis matrix D (MP Bio) and lysed in TRIzol reagent (Invitrogen) for further RNA extraction/purification, utilizing the RNeasy Lipid Tissue Mini Kit (Qiagen). 1 μg total RNA was reverse transcribed using a high-capacity cDNA reverse transcription kit. cDNA was mixed with TaqMan Fast Master Mix (Thermo Fisher) following the manufacturer's recommendations plus selected Taqman probes for gene detection. FAM-labeled probes: *Il6*, *tnfa*, and *Tgm2*. VIC-labeled *Gapdh*, *18s,* or *Actb* were used to normalize gene expression and measure fold changes among groups.

### Total transglutaminase enzymatic activity
Fresh eWAT (100 mg) was incubated in Nonidet P-40 lysis buffer with protease inhibitor cocktail and EDTA for 30 min on ice before high-speed centrifugation. Protein lysates were then used for a colorimetric transglutaminase activity assay following manufacturer's protocol (Abcam; Cat#ab204700) and acquired using a Biotek Synergy HT microplate reader.

### Peptide blocking assay
Anti-TGM2 antibody (Thermo Fisher; Cat# MA5-12739) specificity was tested by incubating anti-TGM2 antibody with native TGM2 protein (R&D Systems; Cat# 5418-TG-010) at a 1:1 (10 ug) ratio for 2 h at room temperature to allow antibody blocking. Native TGM2 protein was then run in two lanes on a 4–12% acrylamide gel, dry-transferred into a nitrocellulose membrane, and incubated with 2% BSA. Next, the membrane was cut into two. One portion was incubated with blocked TGM2 antibody, while the other portion was incubated with control anti-TGM2 antibody not incubated for blocking. Membranes were washed 3x for 5 min in TBST and further incubated in secondary antibody for 30 min at room temperature prior to TBST washing and detected with LICOR.

### Western blot
Lysates were incubated in Nonidet P-40 lysis buffer with protease inhibitor cocktail, EDTA and 100 mg of AT for 30 min on ice before high-speed centrifugation. Lysates were run on 15% gels using vertical gel electrophoresis. Protein content was transferred to nitrocellulose membranes using wet tank or iBlot™ 2 transfer method. β-ACTIN or VINCULIN served as loading controls. After primary antibody staining, which included TGM2 diluted at 1:1000 in TBST (Thermo Fisher; Cat# MA5-12739) and β-ACTIN (Abcam; Cat#ab8229) diluted at 1:5000 in TBST, or VINCULIN (Abcam; Cat#EPR8185), membranes were washed in TBST 3x for 5 min and followed by incubation in secondary antibodies, such as Goat-anti-Mouse conjugated to IRDye680 (LI-COR; Cat# 926-68070), Donkey anti-Goat conjugated to IRDye800 (LI-COR; Cat# 926-32214), or Goat anti-Rabbit IRDye 800LT (LI-COR; Cat#926-32211) diluted to 1:5000 in TBST, which were used for visualization/quantification using the Odyssey imaging Empiria Studio software (LI-COR, Lincoln, NE, USA).

### Tissue histology and fluorescence microscopy
Bone marrow macrophages (BMMs) or eWAT from CD or HFD mice was fixed in 4% paraformaldehyde overnight at +4 °C. Next, samples were placed in a 10% sucrose gradient, embedded in OCT and sectioned at 25 um at -27 °C. Sections were placed on glass slides and air-dried. AT or BMMs (as needed for each experiment) were washed with PBS and permeabilized with 0.1% triton-x for 1 h. Tissue sections were then

incubated in primary antibodies targeting ADRP/Perilipin 2 conjugated to CoraLite®594 (Thermo Fisher; Cat# CL59415294100UL) diluted to 1:200 in PBS, primary unconjugated TGM2 (Thermo Fisher; Cat# MA5-12739) diluted to 1:250 in PBS and further targeted by secondary Goat anti-Mouse IgG, DyLight™ 350 (Thermo Fisher; Cat#62271), F4/80 antibody conjugated to Alexa488, or (Biolegend; Cat#123119) diluted to 1:500 in PBS. Samples were washed in 0.1% triton-x 3x for 5 min IFNy Alexa Fluor® 647 (Biolegend; Cat#505816) and further co-stained with DAPI (Thermo Fisher; Cat# AMEP4650) diluted to 1:100 in PBS and incubated for 5 min prior to final washing procedure. Samples were rinsed with distilled water and air-dried prior to mounting with cover-slips for imaging. eWAT from pLV.Control or pLV.CD11b-Cre treated mice were rinsed in PBS, fixed in 4% paraformaldehyde overnight at +4 °C and rinsed in PBS. Samples were embedded in paraffin and sectioned with a microtome. Samples were then deparaffinated in xylene for 3 min. Tissues were stained with Hematoxylin and Eosin (H&E) by American Histolabs company (Gaithersburg, MD). Images were analyzed in ImageJ using the Adiposoft plugin for quantification of adipocyte number and size[33].

### BMMs and polarization
Bone marrow cells were cultured in DMEM supplemented with 10% fetal bovine serum, 2 mM L-glutamine, 100 U/ml penicillin/streptomycin, and 20 ng/ml M-CSF (for macrophage differentiation) for 7 days in culture. On day 7 of differentiation, cells were stimulated following Peprotech dose recommendations, as follows: M1 = 50 ng/mL IFNy and 10 ng/mL LPS in DMEM and M2 = 20 ng/mL IL-4 in DMEM. Cells were assessed 24 h post-polarization.

### CRISPR silencing
Day 7 BMMs were seeded at 70% confluency for transfection. Next, Cas9 nuclease/sgTgm2 (primary Sequence: CCGGCTGACTCTGTACTTCG; Sequence_B: GCACCAGTTTCTCTTGGCAT; and Sequence_C: CTCACTGTCTGACAATGTGG), or negative control solution were mixed with Cas9 Plus Reagent and further diluted in Opti-MEM I Medium. Cas9 nuclease/sgTgm2 or negative control transfection reagent complex was made by incubating for 10 min at RT and added to cells. Cells were incubated for 2–3 days at 37 °C and then rinsed in PBS to prepare for analyses/experiments.

### In vitro leukocyte co-culture with Tgm2-CRISPR silenced BMMs
$1.2*10^6$ cells derived from 10-week-old mice's eWAT-derived SVF were plated into the bottom portion of a 6-transwell plate (Fisher Scientific; Cat#07-200-165). Tgm2-CRISPR silenced (sgTGM2), or controls (sgCtrl) were pipetted onto the transwell insert located on the top portion of the plate. Cells were incubated for 24 h, further activated with the addition of αCD3/CD28 Dynabeads at 1:4 bead/cell ratio to the SVF and allowed to incubate for 3 days. Supernatants and co-cultured SVF were harvested for analyses.

### eWAT snRNAseq secondary analysis
We downloaded processed Seurat objects from Single Cell Portal accession ID SCP1179 generated by Sarvari et al. 2021[34] containing data from 8355 immune cells ('eWAT_Immune'). 309 out of 8355 cells (3.69%) expressing *Tgm2* were labeled as *Tgm2+*. Cells with zero *Tgm2* expression were labeled as *Tgm2-*. We then performed differential expression comparing *Tgm2 +* cells to *Tgm2-* cells using the Wilcoxon rank-sum test via the FindMarkers function of the Seurat R package v4.3.0.

### Flow cytometry and AT cells sorting
Stromal vascular fraction (SVF) was harvested from eWAT, as described[35] and utilized for: 1) staining of F4/80 + TGM2 ATMs, 2) co-culture studies or 3) AT cell sorting. For staining approach of SVF, co-cultured or rTGM2-treated SVF, cells were harvested and incubated in Fc block Anti-Mouse CD16/CD32 (BD Biosciences; Cat#553141) 1:50 for 5 min on ice and

washed 1x in FACS. Next, cells were incubated in primary conjugated antibodies diluted to 1:100 in FACS, unless otherwise specified for 20 min at 4 °C, as follows: MHC Class II-PerCp-eFluor710 (Thermo Fisher; Cat#46-5321-82), F4/80-PE Cy.7 (Biolegend; Cat#123113), CD206-Brilliant Violet 421 (Biolegend; Cat#141717), CD11c-Alexa Fluor 594 (Biolegend; Cat#117346), TCRβ-PE Cy.7 (Biolegend; Cat#109222), CD45- Alexa Fluor 660 (Thermo Fisher; Cat#606-0451-82), CD4-Alexa 488 (Biolegend; Cat#100425), CD25-PE Cy.5 (Biolegend; Cat#102010), IFNy-PE/Dazzle (Biolegend; Cat#505846), IL-10- APC Cy.7 (Biolegend; Cat#5050335), IL-10- BV421 (Biolegend; Cat#505022), TCRβ- PercP Cy5.5 (Biolegend; Cat#109227), IFNy-BV421 (Biolegend; Cat#505022).

For staining of eWAT-derived SVF ATMs sorting: Panel 1 (Used for protein lysate): Propidium Iodide (Thermo Fisher; Cat# P1304MP), CD45-Alexa Fluor 660 (Thermo Fisher; Cat#606-0451-82), MHC Class II – Brilliant Violet 650 (Biolegend; Cat#107641), F4/80-PE/Cy.7 (Biolegend; Cat#123113), CD11b-PerCP/Cyanine5.5 (Biolegend; Cat#101227), CD11c-Brilliant Violet 421 (Biolegend; Cat#117329). Cells were collected in PBS and further lysed in Nonidet P-40 lysis buffer with protease inhibitor cocktail, EDTA for 1 h on ice before high-speed centrifugation; supernatant was collected as protein lysate. Panel 2 (Used for RNA isolation): Live/Dead-V450 (eBioscience; Cat#65-0863-14), CD45-AFF660 (Fisher scientific; Cat#606-0451-82), MHC Class II- BV510 (Biolegend; Cat#107641), F4/80-PE (Biolegend; Cat#111603), CD14-APCCy.7 (Biolegend; Cat#123317), TCRβ-PECy.7 (Biolegend; Cat#109222), CD31-PerCPCy5.5 (Biolegend; Cat#160206), CD140a-BV605 (Biolegend; Cat#135916); cells were sorted directly into TRIzol reagent for subsequent RNA isolation.

For staining of bone marrow macrophages: CD206-BV605 (Biolegend; Cat# C068C2), IL-10-APC Cy.7 (Biolegend; Cat#5050335), CD64-FITC (Thermo Fisher; Cat#MA5-46784), MHC Class II-Brilliant Violet 510 (Biolegend; Cat#107641), F4/80- Alexa Fluor 594 (Biolegend; Cat#123140). Cells were then washed in 100 ul FACS 3x and re-suspended in 100 uL 4% PFA for 30 min on ice. Next, cells were permeabilized with 0.2% saponin and further resuspended in primary unconjugated TGM2 antibody (Thermo Fisher; Cat# MA5-12739) 1:500 at 2ug/mL or Mouse IgG1 kappa isotype control (eBioscience; Cat#14-4714-82) 1:500 at 2ug/mL overnight at 4̊ C. Cells were then washed in 0.2% saponin 3x and incubated for 30 min on ice with secondary antibody targeting TGM2 or isotype control, Goat anti-Mouse, Alexa Fluor 488 (Thermo Fisher; Cat# A-21238) diluted to 1:5000 in 0.2% saponin. Cells were washed and re-suspended in FACS buffer for data acquisition. Flow cytometric data were analyzed using FlowJo software (v10.8.1) and represented as cells% and/or cells/g, as described. Mouse IgG1 kappa Isotype Control was used to determine TGM2 staining gating strategy.

### ELISA/LEGENDplex
Supernatant was harvested from cell cultured BMMs and evaluated for TGM2 (RayBiotech; Cat#ELM-TGM2-1). BMMs were stimulated with 10 ng/mL LPS for 24 h in vitro for detection of IL-10 concentration (Biolegend; Cat# 431411). Multiparametric LEGENDplex kit was used to detect IFNy, TNFα and IL-6, MCP-1, IL-10, IL-17A, IFNy, TNFα, IL-23, IL-27 or IFNβ in both harvested supernatant and mouse serum (Biolegend; 740150).

### In vitro TGM2 treatments
$4*10^6$ eWAT-derived SVF cells or splenocytes were plated into a 96-U-bottom well plate and incubated with recombinant TGM2 (rTGM2) or vehicle control at 1ug/mL and allowed to incubate for 24 h. Cells were then stimulated using αCD3/CD28 Dynabeads (Thermofisher) and allowed to incubate for 3 days. Cells were harvested, washed, and analyzed by flow cytometry.

### Glucose tolerance (GTT) and Insulin tolerance (ITT) tests
Glucose tolerance (GTT) was performed after fasting mice for 16 h and injecting glucose at 2.5 g/kg of body weight. Insulin tolerance test (ITT) was performed after 4 h fasting conditions, by injecting 0.75 U/kg body weight.

## Statistics and reproducibility

Analysis was performed with GraphPad Prism 9.2.0. Values are represented as mean and SEM. Normality was evaluated for each data set via Shapiro-Wilk test, the Anderson-Darling test, Kolmogorov-Smirnov or D'Agostino & Pearson tests as required by the data. Unpaired Student t test and one-way ANOVA parametric tests were used for normally distributed data. Mann-Whitney non-parametric tests were employed for data not normally distributed. Repeated measures Two-way ANOVA with post-hoc t tests was used for weekly body weight monitoring, GTT and ITT assessments. For all reported data, at least two independent experiments were conducted, and sample sizes were at least five per group.

## Study ethical approval

We have complied with all relevant ethical regulations for animal use. Animal studies followed the standards of humane animal care under protocols approved by the NICHD Animal Care and Use Committee (Animal Study Protocols #21.054 and #24.054).

## Reporting summary

Further information on research design is available in the Nature Portfolio Reporting Summary linked to this article.

## Results

### HFD-induced obese male mice with metabolic dysfunction show increased abundance of TGM2-expressing myeloid cells in epididymal white AT

Mice were fed either CD or HFD for 10 weeks beginning at 6 weeks of age. As expected, measured total body and eWAT weight and blood glucose concentrations were higher in HFD compared to CD-fed control groups (Supplementary Fig. S1A-S1B) along with increased *Il6* and *Tnfa* gene expression in eWAT (Supplementary Fig. S1C), as previously demonstrated[36,37]. HFD mice also showed impaired GTT during glucose and insulin tolerance tests (Supplementary Fig. S1D-S1E), thereby confirming metabolic dysfunction in our diet-induced obese mouse model. To investigate how TGM2 body tissue expression is altered in these HFD mice, we evaluated TGM2 protein expression in metabolically relevant tissues. TGM2 protein expression was significantly increased in the liver, but not in the kidney or pancreas in HFD mice compared to CD controls (Supplementary Fig. S2A). Notably, HFD-treated mice had a two-fold increase in total AT transglutaminase activity as compared to CD mice (Fig. 1A). This was corroborated by observing a 2.5-fold increase in *Tgm2* expression in the eWAT from HFD as measured by qPCR (Fig. 1B) and a significant increase in TGM2 protein expression as assessed by western blot upon normalization to *B*-ACTIN or VINCULIN (Fig. 1C). Thus, there is increased TGM2 activity in the eWAT from HFD male mice (compared to controls) in conjunction with the increased tissue inflammation and metabolic dysfunction associated with obesity.

To determine the source of AT TGM2 in HFD-induced obesity, a secondary analysis of snRNAseq eWAT published data by Sarvari et al.[34], showed an increased abundance of *Tgm2* expressing cells belonging to the immune cell fraction in HFD mice compared to CD group (Fig. 1D). Importantly, when narrowing down the analysis to focus on the immune cell populations, ATMs were the subset that showed a higher number of cells positive for *Tgm2* in the HFD group compared to CD control (Fig. 1E). Next, fluorescence microscopy corroborated TGM2 co-localization with F4/80+ macrophages in eWAT in both CD and HFD mice (Fig. 1F). Corroborative flow cytometric assessment in SVF from CD or HFD mice (gating strategy shown in Supplementary Fig. S2B) showed an increased abundance of F4/80 + ATMs (Fig. 1G), that co-expressed TGM2 (Fig. 1H) (as tested with a specific TGM2 antibody shown in Supplementary Fig. S2C-S2E). There was an increased number of TGM2 + ATMs in the eWAT from HFD mice compared to CD controls.

Furthermore, to investigate if intracellular levels of *Tgm2* differed in CD vs. HFD eWAT-derived cell populations, eWAT-derived SVF cells were sorted for preadipocytes, macrophages, and monocytes (gating strategy shown in

Supplementary Fig. S2F). No significant differences in intracellular *Tgm2* expression by qPCR analysis were observed in CD vs. HFD preadipocytes or macrophages, while a significant increase in intracellular *Tgm2* was shown in HFD eWAT-SVF sorted monocytes, compared to CD controls (Fig. 1I). This indicates that most of the increased *Tgm2* in the eWAT from HFD male mice emanates from an increased number of *Tgm2* + ATMs and the presence of monocytes expressing high intracellular levels of *Tgm2* compared to CD controls. Lastly, to investigate the inflammatory profile of *Tgm2*- vs. *Tgm2* + eWAT ATMs in HFD male mice, we re-evaluated the snRNAseq data for CD and HFD mice published by Sarvari et al.[34], focusing on the differential gene expression profile of the two populations. These data analysis found that cathepsin D (*Ctsd*), cathepsin L (*Ctsl*), and Matrix metalloproteinase-12 (*Mmp12*) were increased in the *Tgm2* + ATMs compared to the *Tgm2*- ATMs, while solute carrier family 9 member A9 (*Slc9a9*) was reduced (Fig. 1J).

Altogether, these data point to the increased abundance of TGM2+ AT myeloid cells in HFD as a candidate player in regulating AT inflammation during obesity scenarios with metabolic dysfunction.

### *Tgm2* CRISPR silencing in BMMs results in increased pro-inflammatory macrophage profile in vitro

To determine if TGM2 expression in macrophages is a key modulator of their inflammatory profile, we employed CRISPR silencing of *Tgm2* expression (sgTGM2) or scramble negative control (sgCtrl) in BMMs undergoing M1 (generally pro-inflammatory) or M2 (generally anti-inflammatory) polarization. We used BMMs from CD mice because no significant differences in TGM2 expression in differentiated BMMs from CD vs. HFD mice were observed by fluorescence microscopy (Supplementary Fig. S3A), flow cytometric or qPCR analysis (Supplementary Fig. S3B-S3F), confirming previously discussed data showing that although *Tgm2* intracellular levels remain equal in macrophages from CD vs. HFD treatment groups, observed differences in total tissue TGM2 may be due to the increased number of TGM2 + macrophages.

Successful silencing and reduced constitutive secreted TGM2 in the sgTGM2 compared to sgCtrl was demonstrated by western blot analysis (Fig. 2A), and ELISA (Fig. 2B), respectively. No significant changes were observed in total transglutaminase activity (Fig. 2C), indicating potential unimpeded expression of other transglutaminases.

Next, we evaluated the secreted inflammatory cytokine profile from CRISPR-targeted LPS-stimulated macrophages in the supernatant to determine *the effects of Tgm2* silencing on inflammation. Results revealed a significant increase in IFN-γ release in the sgTGM2 BMMs compared to sgCtrl. No significant changes were observed in other assessed cytokines, such as TNFα or IL-6 (Fig. 2D). To further determine if *Tgm2* silencing in macrophages affects their polarization profile, we next stimulated CRISPR targeted BMMs with M1 (+LPS, +IFNy), M2 (+IL-4) or M0 (Vehicle) polarizing cytokines. Cells were analyzed by flow cytometric analyses for expression of M1 or M2 polarization markers by focusing on the TGM2+ , CD206+, IL-10+ , CD64+ , MHC Class II+ or F4/80+ cell% populations upon M0, M1, or M2 polarization stimuli. As expected, evaluation of polarizing conditions in BMMs resulted in M2 polarization to induce CD206, while M1 polarization induced CD64 and MHC Class II and downregulated IL-10. Interestingly, TGM2 was significantly suppressed in M1 polarization when compared to M0 basal conditions in sgCtrl-treated BMMs (Supplementary Fig. S4A), suggesting a negative regulation of TGM2 under acute pro-inflammatory scenarios.

We further analyzed flow cytometric data in order to determine if absence of TGM2 during M0/M1/M2 polarizing conditions would impair macrophage polarization capacities. Results showed the expected significant reduction in TGM2 + BMMs but also a significant reduction in IL-10+ BMMs and a significant increase in CD64 and MHC Class II markers under M0 stimuli (Fig. 2E), while no differences were observed in CD206 when compared to sgCtrl BMMs. M1 polarization in sgTGM2 BMMs (Fig. 2F) resulted in the expected significant decrease in TGM2, but also reduced CD206 + , and IL-10+ BMMs, while MHC Class II was

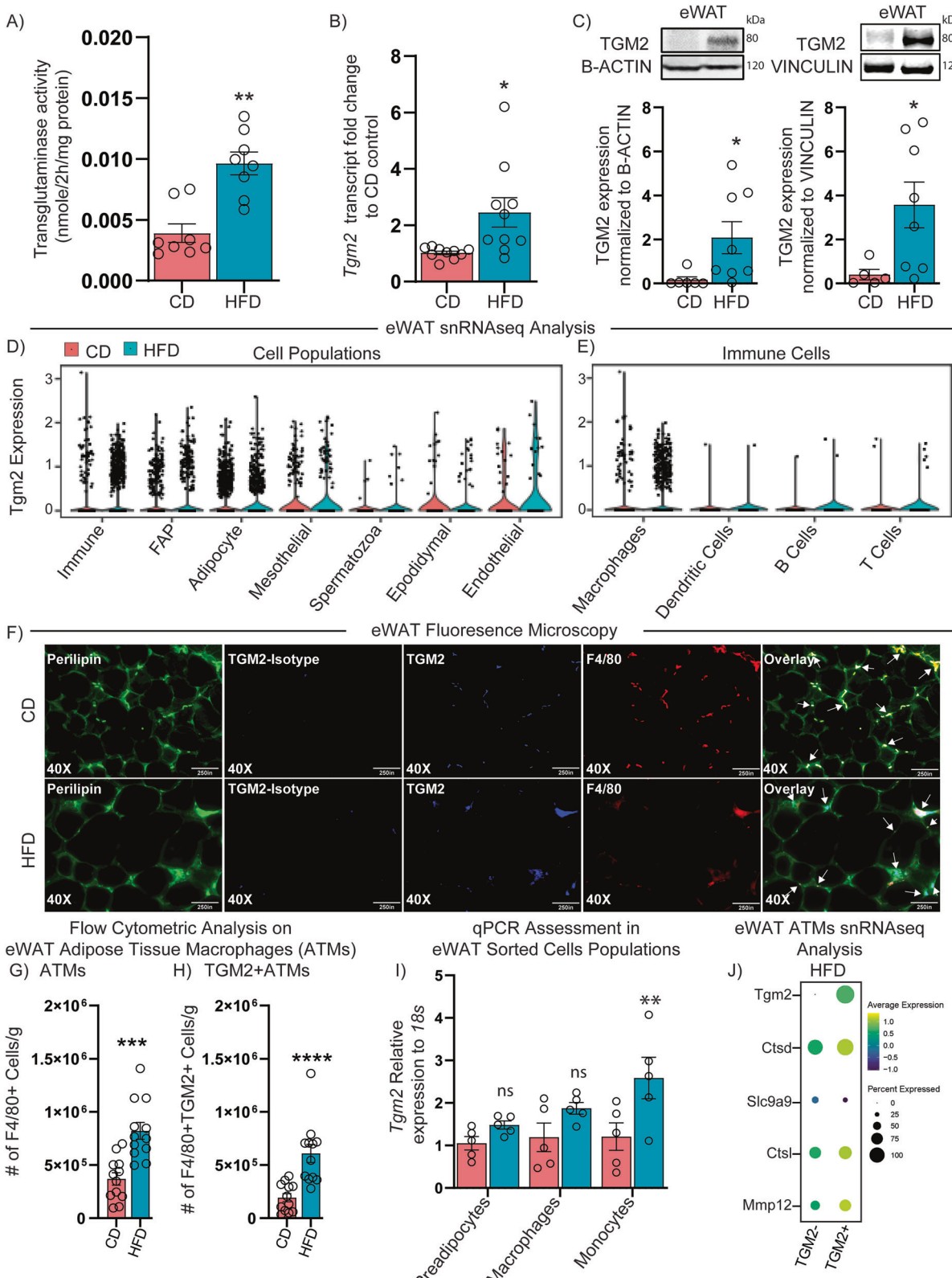

upregulated in sgTGM2 BMMs. No differences were seen in CD64, or F4/80 markers in sgTGM2 BMMs when compared to sgCtrl. Lastly, M2 polarization of *Tgm2* silenced BMMs (Fig. 2G) resulted in the expected significant decrease of TGM2, but also decreases of CD206, IL-10, and F4/80 markers, while MHC Class II was upregulated, and no significant changes were observed in CD64, when compared to sgCtrl BMMs,

indicating a potential polarizing role of TGM2 expression in BMMs under M0, M1, and M2 inflammatory settings. Lastly, two additional sgTGM2 CRISPR sequences targeting *Tgm2* silencing in BMMs corroborated decreased TGM2 expression as well as lower IL-10, increased MHC Class II, and a trend towards increasing CD64 upon *Tgm2* silencing under M0 basal conditions (Supplementary Fig. S4B–S4C), thereby

**Fig. 1 | TGM2 co-expression with adipose tissue macrophages (ATMs) in diet-induced obesity.** C57BL/6 J mice were placed on chow diet (CD) or 60% high fat diet (HFD) at 6 weeks of age and remained on diet treatment until 16 weeks of age. **A** Total transglutaminase activity assay in eWAT from CD ($n = 8$) vs. HFD ($n = 8$) mice. **B** qPCR evaluation of *Tgm2* mRNA expression in eWAT from CD ($n = 10$) or HFD ($n = 10$) mice. **C** Western blot for TGM2 protein expression in eWAT from CD control ($n = 11$) or HFD-treated mice ($n = 16$) normalized to B-ACTIN or VINCULIN expression. eWAT-derived single-nucleus RNA-seq from C57BL/6 J mice placed on a 60% HFD for 18 weeks ($n = 3$ pooled mice) showing *Tgm2* expression in adipose tissue **D** global tissue populations or **E** immune cells in CD vs. HFD diet-treatment groups, where we observed a greater number of cells in the HFD sub-group as evidenced by the larger number of dots in the violin plot. Next, eWAT-derived from CD ($n = 3$) or HFD-treated mice ($n = 3$) was cryosectioned or paraffin sectioned and fluorescently stained for **F** Perilipin, F4/80 and intracellular TGM2 (or corresponding isotype control) for microscopy analysis. **G** CD ($n = 12$) vs. HFD ($n = 12$) eWAT-derived SVF was analyzed by flow cytometry for **H** number of macrophages and **I** Number of macrophages expressing TGM2 in CD vs. HFD diet treatment groups. **I** CD ($n = 5$) vs. HFD ($n = 5$) eWAT-derived SVF sorted pre-adipocytes, macrophages or monocytes evaluated for *Tgm2* expression by qPCR assessment. **J** snRNAseq assessment of 60%HFD *Tgm2*- vs. *Tgm2* + ATMs differential gene expression profile. Data are shown as bar graphs with SEM of five or twelve mice per control or treated group and are representative of two -four independent experiments. For all graphs, data normality status was used to determine statistical analysis. Normal data statistical significance was determined by parametric student's unpaired *t* test, while not normally distributed data were analyzed using non-parametric Mann-Whitney test. ns = $P > 0.05$, *= $P < 0.05$, ** = $P < 0.01$, ***$P < 0.001$.

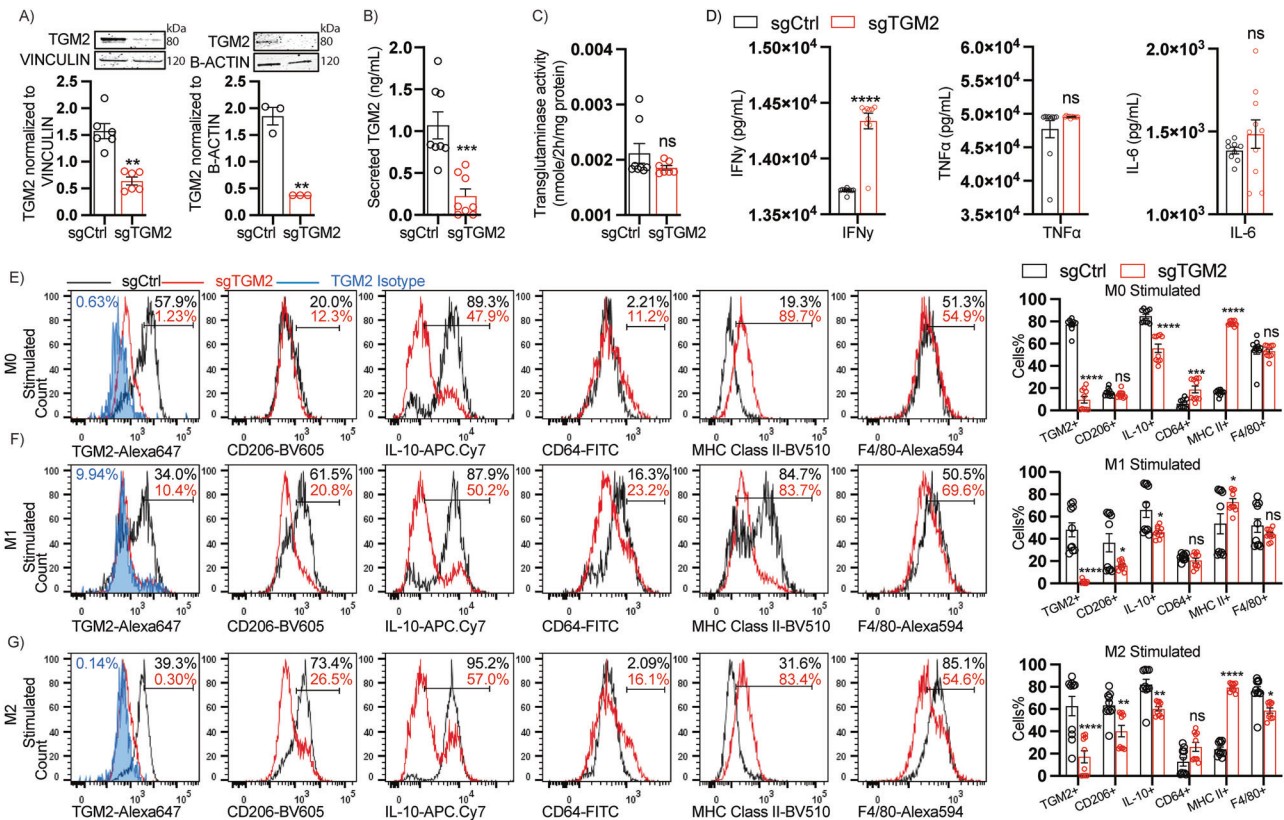

**Fig. 2 | Immunophenotyping of *Tgm2* CRISPR-Silenced BMMs.** Bone marrow was isolated from C57BL/6 J mice and cultured in the presence of M-CSF for BMMs differentiation for 4 days. BMMs were then transfected with CRISPR targeting *Tgm2* and evaluated 72 h post-transfection. **A** Western blot analysis confirming *Tgm2* silencing in sgControl ($n = 9$) or sgTGM2 ($n = 9$) transfected BMMs. **B** Evaluation of TGM2 secretion in CRISPR silenced BMMs ($n = 8$/group), **C** Transglutaminase enzymatic activity in *Tgm2* CRISPR silenced BMMs ($n = 7$/group) and **D** Cytokine profile in harvested supernatant from CRISPR targeted BMMs treated with LPS for 24 h ($n = 9$–10/group). **E** Gating strategy for flow cytometric analysis of CRISPR silenced BMMs ($n = 10$/group). Flow cytometric data analysis from CRISPR silenced BMMs polarized into **E** M0, **F** M1 (50 ng/mL IFNy and 10 ng/mL LPS in DMEM) or **G** M2 (M2 = 20 ng/mL IL-4 in DMEM), polarizing stimuli, on day 7 of differentiation. Cells were assessed 24 h post-polarization. Data are shown as bar graphs with SEM of seven or ten mice per control or treated group and are representative of three independent experiments. For all graphs, data normality status was used to determine statistical analysis. Normal data statistical significance was determined by parametric student's unpaired t-test, while not normally distributed data was analyzed using non-parametric Mann-Whitney test. ns = $P > 0.05$, **=$P < 0.05$, ** =$P < 0.01$, ***$P < 0.001$.

reducing the likelihood that we were observing off-target effects from our employed CRISPR approach.

### In vitro inhibition of TGM2 secretion from BMMs increases pro-inflammatory markers in co-cultured AT leukocytes

TGM2 is localized in the nucleus, plasma membrane, cytosol, mitochondria, and extracellular matrix[14]. Importantly, previous works have determined that TGM2 cellular localization is a critical determinant of its function. For instance, TGM2 secretion into the extracellular environment regulates cellular activities, via Ca2 + dependent transamination[38]. Although previous evidence has led to the belief that TGM2 cannot be secreted in the absence of cellular stress[39,40], studies have shown that overexpression of TGM2 in 3T3 fibroblasts results in successful externalization into the ECM in absence of exogenous stimuli or ongoing cell death[41], making TGM2 secretory mechanisms unelucidated. To investigate if TGM2 serves as a soluble mediator for macrophage crosstalk with AT leukocytes inflammatory profile, sgTGM2 CRISPR silenced (with impaired TGM2 release, as shown in Fig. 2B) or sgCtrl BMMs were co-cultured with

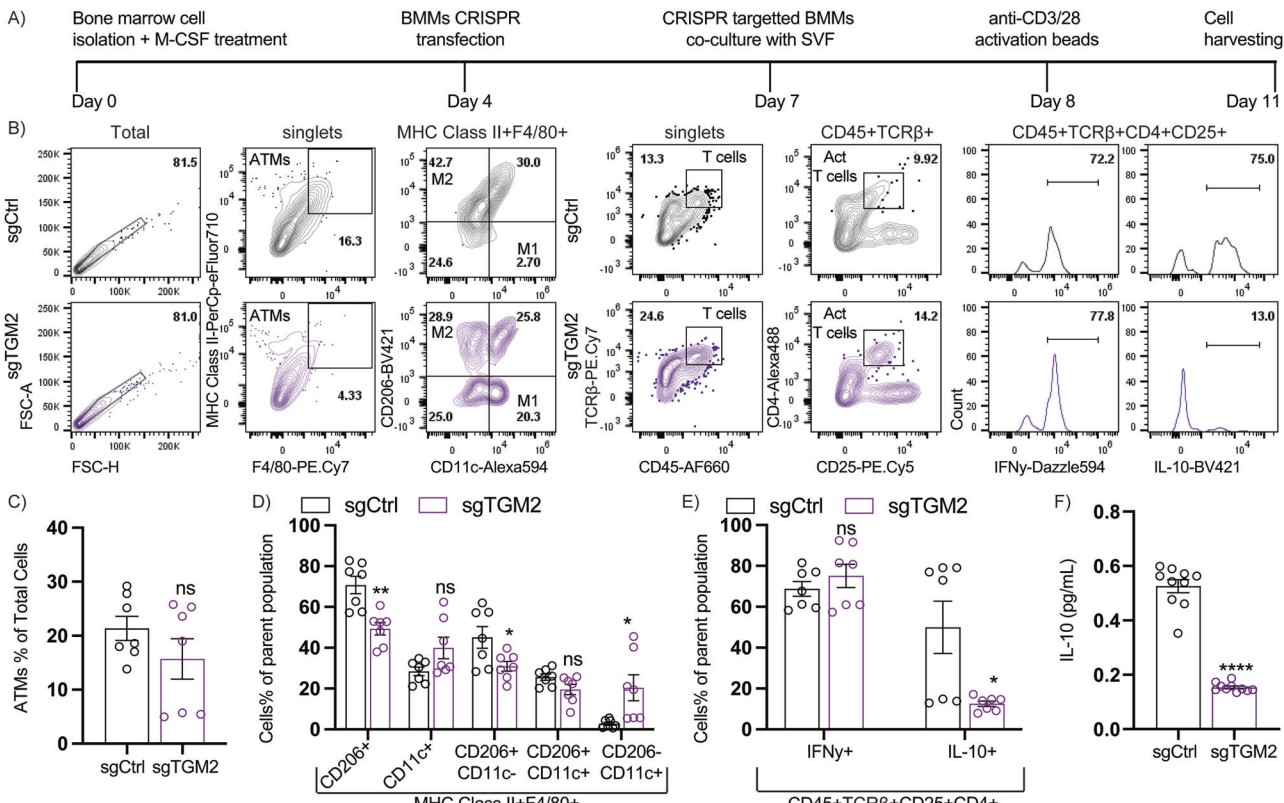

**Fig. 3 | Macrophage-derived TGM2 effects on adipose tissue leukocyte inflammatory profile.** CD C57BL/6 J mice's long bones were used for preparation of bone marrow-derived macrophages (BMDM) ex vivo. Next, bone marrow-derived macrophages were transfected with CRISPR sgTGM2 or sgCtrl to silence *Tgm2* expression on day 4 of the ex vivo culture timeline and co-cultured with eWAT-derived SVF in the presence of T cell activating anti-CD3/28 magnetic beads. **A** Study timeline. **B** Flow cytometric gating strategy and analysis for **C–D** myeloid cells (MHC Class II + F4/80+ gated) and **E** T cells (CD45 + TCRb + CD25 + CD4 + gated) in the co-cultured SVF cells (*n* = 7/group), and **F** ELISA for IL-10 in supernatant from co-cultured sgTGM2 or sgCtrl BMMs and SVF cells (*n* = 10/group). For all graphs, data normality status was used to determine statistical analysis. Normal data statistical significance was determined by parametric student's unpaired *t* test, while not normally distributed data were analyzed using non-parametric Mann-Whitney test. ns = $P > 0.05$, *= $P < 0.05$, ** = $P < 0.01$, ***$P < 0.001$.

epididymal eWAT-derived SVF from CD mice in transwell plates. To stimulate $Ca^{2+}$ - dependent TGM2 secretion/activity in co-cultured BMMs[42], we induced a pro-inflammatory environment (as seen in HFD AT) in the co-cultured cells, via anti-CD3/CD28 activation beads for 72h[43]. Co-cultured SVF cells were then harvested for analysis, as indicated by the study timeline illustrated in Fig. 3A.

We then performed flow cytometric analysis on harvested SVF and evaluated expression of markers associated with inflammation in AT, such as CD11c[44] or CD206[45], IFNy, or IL-10. Gating strategy consisted of evaluating CD45 + MHC Class II + F4/80+ macrophage expression of CD11c vs. CD206 for myeloid cells, while for T cell assessment, gating strategy evaluated CD45 + TCRβ + CD4 + CD25 + T cells' expression of IFNy or IL-10 (gating strategy shown in (Fig. 3B). Results showed no significant differences in total ATMs (MHC Class II + F4/80 + )between sgCtrl and sgTGM2 groups (Fig. 3C), while MHC Class II + F4/80+ gated SVF co-cultured in the presence of sgTGM2 BMMs exhibited significantly reduced CD206+ cell population, CD206 + CD11c- population, and increased CD206-CD11c+ population, while the CD11c+ and CD206+ CD11c+ were not significantly altered (Fig. 3D). This suggests that the absence of secreted TGM2 in co-cultured BMMs significantly reduced anti-inflammatory CD206 marker expression in co-cultured SVF. Furthermore, flow cytometric analysis of CD45+ TCRβ+ CD25+ CD4+ gated AT T cells (ATT), showed no significant changes in the IFNy+ subset, while the IL-10 subset was shown to be significantly reduced in the ATT cells co-cultured with sgTGM2 BMMs compared to sgCtrl (Fig. 3E).

Lastly, to evaluate if TGM2 absence altered anti-inflammatory IL-10 secretion, we examined IL-10 by ELISA in harvested supernatant, finding a significant decrease in IL-10 concentrations in the co-culture system (Fig. 3F); follow-up investigations on IL-10 expression in other components in the co-cultured SVF cells, such as myeloid cells would confirm if secreted IL-10 was also perturbed in other AT leukocytes. Overall, macrophage-derived TGM2 was shown to be a critical factor involved in macrophage-AT immune cell cross talk to maintain anti-inflammatory properties even in the presence of acute pro-inflammatory stimuli.

### rTGM2 protein exposure to AT leukocytes promotes anti-inflammatory IL-10 expression in T cells under homeostatic conditions

To further investigate if TGM2 was a key soluble factor inducing anti-inflammatory properties in co-cultured SVF, we cultured CD or HFD eWAT-derived SVF in the presence of rTGM2 at 1 ug/mL, or vehicle control for 24 h. Next, SVF cells were stimulated 24 h post-rTGM2 exposure with T cell stimulating anti-CD3/CD28 activation beads and remained in culture throughout the next 72 h (as depicted in Fig. 4A). Importantly, effectiveness of T cell activation was confirmed by gating cells based on CD45, CD4, CD25, IL-10 or IFNy marker expression, as depicted in Supplementary Fig. S5A. CD45+ TCRβ+ CD25+ CD4+ IL-10+ cells% (corresponding to activated IL-10 + T cells) were compared in unstimulated vs. anti-CD3/ 28 stimulated vehicle controls in CD or HFD-derived SVF cells (Supplementary Fig. S5B). Interestingly, results showed successful activation only in the CD group, particularly in the IL-10 + T cell marker expression and

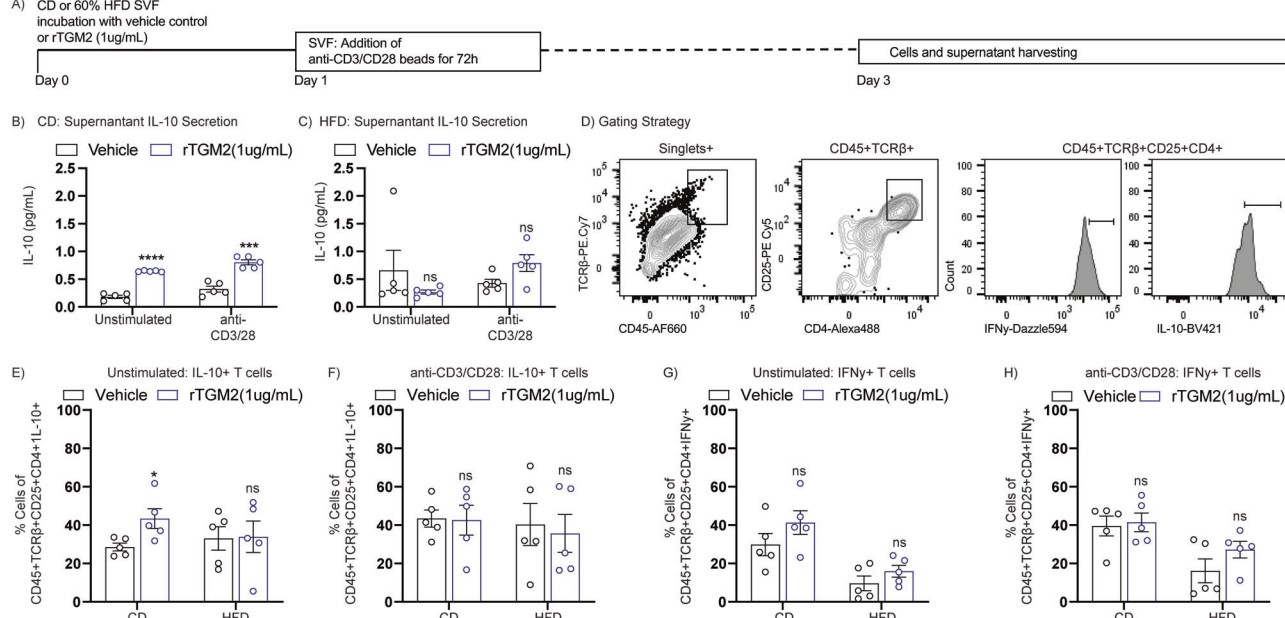

**Fig. 4 | Recombinant TGM2 effects in adipose tissue leukocytes inflammatory profile. A** eWAT SVF cells were harvested from CD C57BL/6 J mice and processed for single cell suspension, followed-up by ACK red blood lysis treatment. Cells were then plated in a 96-U-bottom well plate in the presence of recombinant TGM2 (rTGM2) 1.0 ug/mL, or vehicle control for 24 h. Supernatant was assessed for IL-10 secretion by ELISA in SVF cells collected from **B** CD or **C** HFD treated mice (n = 5/ group). **D** Harvested cells were assessed with flow cytometry analysis in unstimulated or anti-CD3/CD28 stimulated conditions for **E–F** CD45+ TCRb+ CD25+ CD4+

IL-10+ T cell populations, or **G–H** CD45+ TCRb+ CD25+ CD4+ IFNγ+ T cell populations (n = 5/group). Data are shown as bar graphs with SEM of five mice per control or treated group and are representative of two independent experiments. For all graphs, data normality status was used to determine statistical analysis. Normal data statistical significance was determined by parametric student's unpaired t test, while not normally distributed data were analyzed using non-parametric Mann-Whitney test. ns = P > 0.05, *=P < 0.05, ** =P < 0.01, ***P < 0.001.

cytokine release (Supplementary Fig. S5B and S5D). Although trending towards an increase in IFNγ+ T cells, there was no significant differences in unstimulated vs. anti-CD3/28 stimulated groups (Supplementary Fig. S5C), potentially due to a technical difficulty in appropriately detecting sufficient intracellular IFNγ levels post-cell stimulation.

Additionally, the HFD group remained unresponsive to external anti-CD3/28 bead stimulation, attributable to previously demonstrated T cell exhaustion occurring in HFD ATTs[46]. Next, to evaluate if TGM2 protein treatment exerted previously observed anti-inflammatory effects under no external stimuli or anti-CD3/28 stimulation compared to vehicle control, we first evaluated supernatant for IL-10 by ELISA, which showed significantly increased IL-10 secretion in CD SVF treated with rTGM2 compared to vehicle control under anti-CD3/CD28 or unstimulated conditions (Fig. 4B), while no significant differences were observed in the supernatant from HFD SVF cells treated with rTGM2 (Fig. 4C). Next, to investigate T cell profile upon rTGM2 treatment, we performed flow cytometric analysis on SVF gating on CD45+ TCRβ+ CD25+ CD4+IL-10+ or CD45+ TCRβ+ CD25+ CD4+IFNγ+, as depicted in Fig. 4D. Results showed increased IL-10+ T cell subset in the unstimulated CD SVF cells treated with rTGM2, while no significant changes were observed in the HFD SVF cells (Fig. 4E).

Interestingly, flow cytometric analysis on the anti-CD3/CD28 stimulated rTGM2 treated SVF cells, showed no significant differences in IL-10+ T cells in the CD or HFD groups (Fig. 4F), suggesting insufficient ability of TGM2 in the absence of potential complementary macrophage-derived secreted cytokines to exert significant anti-inflammatory effects in CD4 + T cells under pro-inflammatory environments. We then aimed to evaluate if supplementation of TGM2 to SVF cells could potentially prevent expression of pro-inflammatory IFNγ when activating SVF T cells via anti-CD3/CD28 stimuli. Data showed no significant differences in CD45+ TCRβ+ CD25+ IFNγ+ T cells from CD nor HFD treated with rTGM2 in comparison to vehicle controls (Fig. 4G). Altogether, results suggest that TGM2 + macrophages may use secreted TGM2 to communicate with ATMs and T cells to stimulate anti-

inflammatory profiles, but that these effects have much less impact during chronic pro-inflammatory states associated with obesity onset and progression.

## In vivo CD11b myeloid cell silencing of *Tgm2* results in increased tissue and systemic inflammation, accompanied by increased diet-induced obesity and IR

To continue investigating the effects of TGM2 expression in myeloid cells on diet-induced obesity inflammation and metabolic dysfunction, we utilized a lentivirus approach to induce silencing of *Tgm2* in CD11b+ myeloid cells. Briefly, we employed a lentivirus control (pLV.Control) carrying human EF1A promoter upstream of mCherry and EGFP reporter downstream of ubiquitously expressed CMV promoter, as well as a lentivirus to silence *Tgm2* in CD11b+ cells (pLV.CD11b-Cre) via expression of CD11b promoter upstream of Cre recombinase and EGFP reporter downstream of CMV promoter (Fig. 5A). Lentiviruses were intraperitoneally injected into *Tgm2* floxed homozygous mice at 5 weeks of age and placed on a 60% HFD one-week post-injections and monitored for body weight changes on a weekly basis. Additional assessments included GTT and ITT, whole body composition by DEXA scan and flow cytometric analysis of AT SVF immune cell profile, as denoted in the study timeline (Fig. 5B).

We corroborated that silencing of *Tgm2* in CD11b+ myeloid cells within the AT resulted in AT immune cell profile changes via flow cytometric assessment in the SVF, as shown in gating strategy (Fig. 5C). There was a significant reduction in TGM2+ ATMs (CD45+ CD11b+ GFP+ F4/80+ MHC II+ TGM2+), a significant increase in total leukocytes (CD45+) and in MHC IIhi+ATMs (CD45+ CD11b+GFP+ F4/80+ MHC IIhi+) in the pLV.CD11b-cre injected mice, compared to pLV.Control, while no significant differences were observed in the MHCIIlo+ATMs (CD45+ CD11b+ GFP+ F4/80+ MHC IIlo+), IL-10+ATMs (CD45+ CD11b+GFP+ F4/80+ MHC II+ IL-10+), or ATT (CD45+ CD11b-GFP-F4/80-MHC IIhi-TCRb+) between groups (Fig. 5D). Our data show that about 60% of the macrophages were

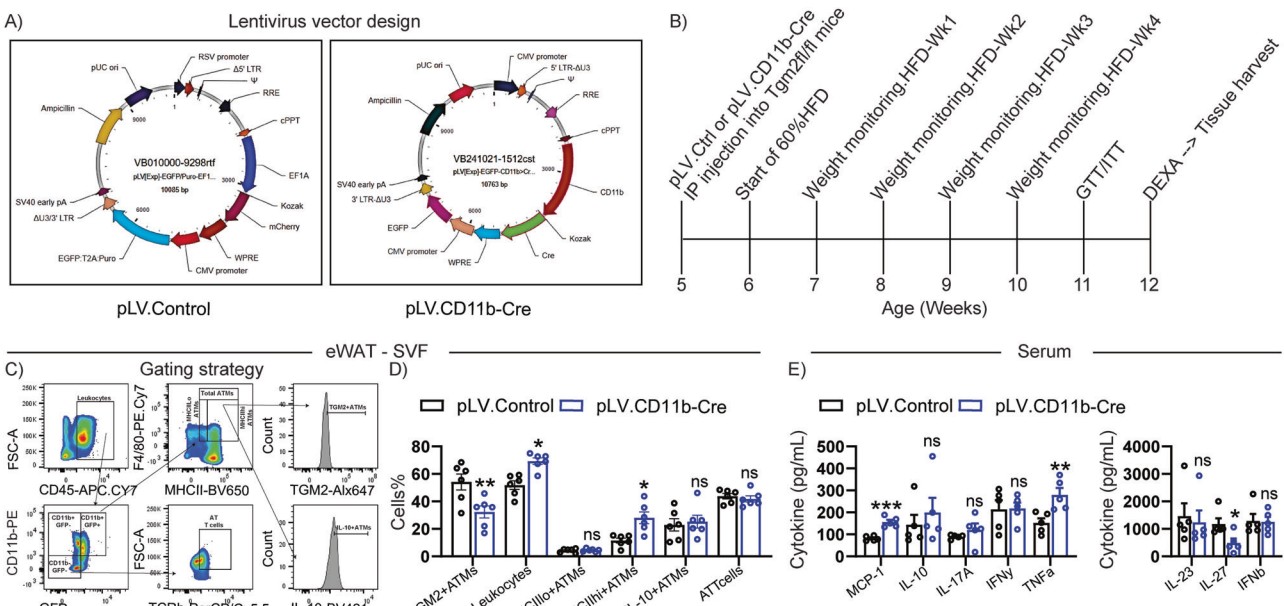

**Fig. 5 | In vivo studies of effects of lentivirus carrying CD11b promoter upstream of Cre recombinase to silence *Tgm2* in TGM2fl/fl male mice.** *Tgm2* floxed mice (B6.129S1-*Tgm2^{tm1Rmgr}*/J) were injected at 5 weeks of age with **A** pLV.Control or pLV.CD11b-Cre lentiviruses, followed by CD or 60% HFD treatment and further metabolic evaluations. **B** Timeline for procedures. **C** Flow cytometric gating strategy employed for eWAT-derived SVF from pLV.Control vs. pLV.CD11b-Cre treated mice and **D** subsequent analysis looking at % of TGM2+ATMs (CD45+ CD11b+GFP+ F4/80+ MHCII+), Leukocytes (CD45+), MHCIIlo+ATMs (CD45+ CD11b+GFP+ F4/80+ MHCII-), MHCIIhi+ATMs (CD45+ CD11b+ GFP+ F4/80+MHCIIhi+), IL-10+ATMs (CD45+ CD11b+GFP+ F4/80+

MHCII+ IL-10+ ), and AT T cells (CD45+ CD11b+GFP+ F4/80-MHCII-TCRb +) (*n* = 5/group). **E** LegendPlex assessment of cytokine inflammatory profile in serum from pLV.Control vs. pLV.CD11b-Cre injected mice (*n* = 5/group). Data are shown as bar graphs with SEM of five mice per control or treated group and are representative of two independent experiments. Graphs were statistically analyzed by Two-way ANOVA for flow cytometric populations. For all bar graphs, data normality status was used to determine statistical analysis. Normal data statistical significance was determined by parametric student's unpaired *t* test, while not normally distributed data were analyzed using non-parametric Mann-Whitney test. ns = $P > 0.05$, *=$P < 0.05$, ** =$P < 0.01$, ***$P < 0.001$.

successfully silenced for *Tgm2* upon lentivirus incorporation and that this reduction of TGM2+ expressing ATMs leads to an increase in pro-inflammatory cell profile within the AT of diet-induced obese mice. Next, assessment of serum from pLV.Control vs. pLV.CD11b-Cre injected mice showed an increased inflammatory cytokine profile, as shown by significantly higher levels of MCP-1 and TNFα, while no significant differences were observed in IL-10, IL-17A, IFNy, IL-23 or IFNβ between the groups (Fig. 5E). Interestingly, data showed a significant downregulation of circulating IL-27 in the pLV.CD11b-Cre injected mice compared to pLV.Control, which despite possessing both pro- and anti-inflammatory properties[47], has been proven to positively modulate weight gain, adiposity, and IR[48]. Altogether, these data suggest that silencing of TGM2 in CD11b+ myeloid AT cells result in increased inflammation in diet-induced obese male mice.

Concomitantly, to evaluate the metabolic health effects of *Tgm2* silencing in CD11b myeloid cells for both male and female mice, we evaluated body weight changes on a weekly basis, finding a significant increase in whole body weight from male mice injected with pLV.CD11b-Cre compared to pLV.Control mice (Fig. 6A). Furthermore, when evaluating whole body composition by DEXA scanning, results showed no significant changes in lean mass, while fat mass was increased in the pLV.CD11b-Cre male (Fig. 6C–D) and female (Supplementary Fig. S6C-D) mice. To evaluate if changes in eWAT mass were contributing to observed increased DEXA fat mass, we assessed gross eWAT and liver tissue weight post-mortem, where results showed increased eWAT tissue weight in the pLV.CD11b-Cre male (Fig. 6E) and female mice (Supplementary Fig. S6E) compared to pLV.Control, while no significant weight differences were apparent in the liver (Fig. 6F; Supplementary Fig. S6F). Further histological assessment of eWAT stained for H&E (Fig. 6G) showed a significant increase in adipocyte size in the pLV.CD11b-Cre compared to pLV.Control male mice (Fig. 6H). We saw no significant differences in glucose sensitivity (Fig. 6I) in pLV.CD11b-Cre compared to pLV.Control male mice, while insulin

sensitivity was significantly impaired in pLV.CD11b-Cre mice compared to pLV.Control (Fig. 6J). Female metabolic assessments mirrored findings in male data (Supp Fig. S6I-J), indicating that *Tgm2* silencing in CD11b+ myeloid cells play a significant role in the pathophysiology of obesity, suggesting an absence of important sexual dimorphisms. Altogether, these data indicate that TGM2 expression in ATMs is an essential player modulating AT inflammation associated with metabolic dysfunction.

## Discussion

These findings suggest that macrophage derived TGM2 has an important anti-inflammatory role related to AT inflammation. First, results corroborate studies showing TGM2 to be elevated in obesity[49,50]. Second, the relationship between increased TGM2 and ATMs in male diet-induced obese mice showed TGM2 to be largely co-localized in the F4/80+ ATMs. Third, using CRISPR *Tgm2* silencing and administration of TGM2, we demonstrated its role as an intrinsic and paracrine modulator of inflammation in AT leukocytes. Lastly, in vivo silencing of *Tgm2* in CD11b + myeloid cell populations resulted in augmented pro-inflammatory responses at both the AT and systemic levels, ultimately causing increased obesity and IR.

Our secondary analysis of eWAT snRNAseq data showed *Tgm2* expression in FAP, adipocyte, and endothelial cell populations from both CD and HFD eWAT, indicating that ATMs are not the sole contributor of TGM2 in AT during HFD. However, it was the *Tgm2*+ ATMs that showed an obvious increase in cell numbers in the HFD group compared to CD control, as corroborated by data available in the open-access web portal describing immune cell phenotypes from weight loss and subsequent weight regain in mice eWAT[51]. This suggests that the increased macrophage population in eWAT from HFD significantly augments the overall expression of TGM2 at the tissue level. Additionally, our data also showed increased expression of *Tgm2* in the liver from HFD male mice compared to CD controls, suggesting that TGM2 in liver tissue could also contribute to

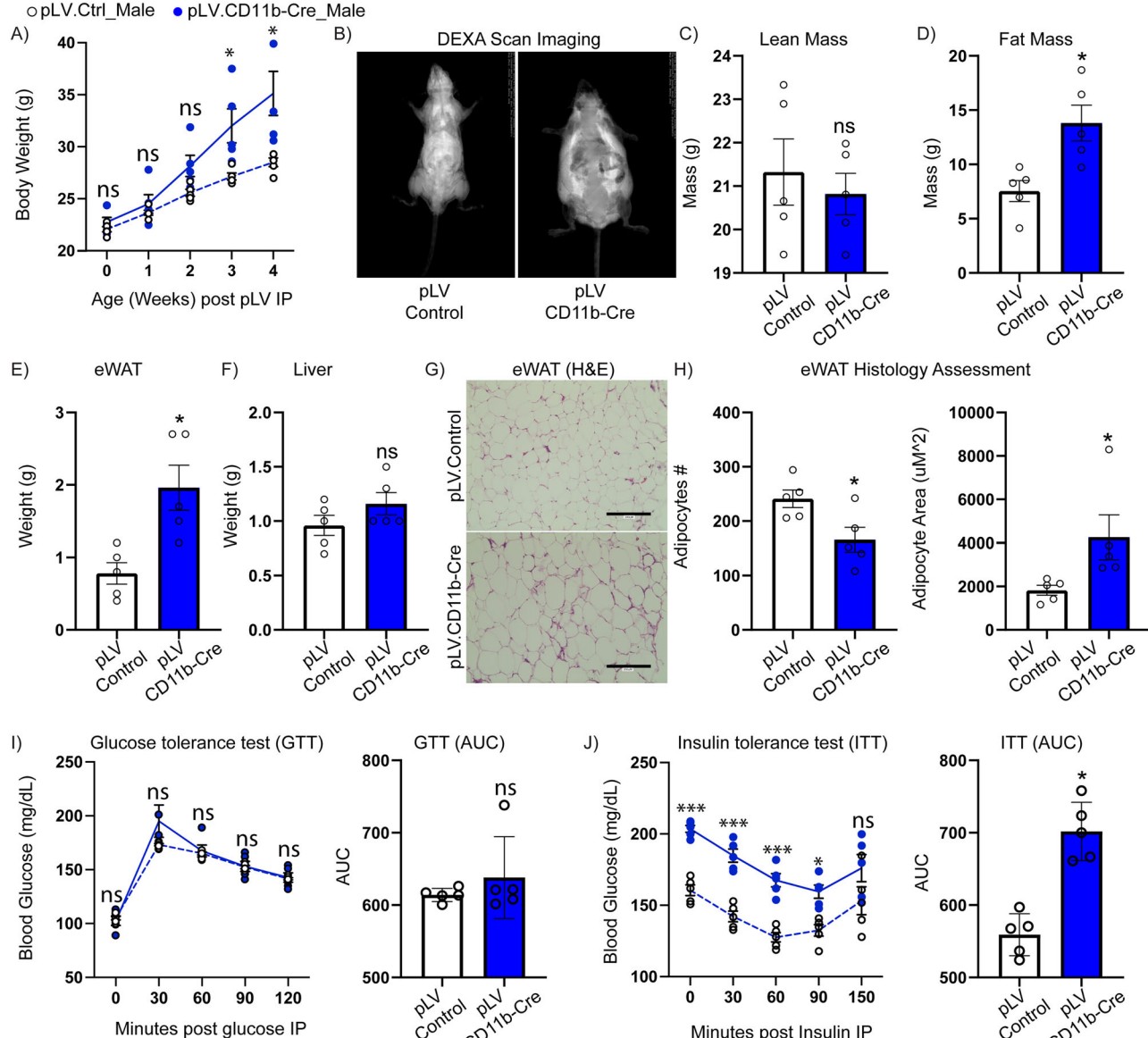

**Fig. 6 | In vivo silencing of *Tgm2* in myeloid cells: effects on male mouse metabolic health.** *Tgm2* male floxed homozygous mice injected with pLV.Control or pLV.CD11b-Cre lentiviruses at 5 weeks of age were placed on a 60%HFD at 6 weeks of age. **A** weekly weight monitoring (*n* = 5/group). **B** DEXA scanning evaluating changes in **C** lean and **D** fat mass (*n* = 5/group). **E** gross eWAT and **F** liver weights post-study termination (*n* = 5/group). **G** eWAT H&E histology and **H** histological analysis evaluating number of adipocytes and adipocyte area in pLV.Control vs. pLV.CD11b-cre treated mice (*n* = 5/group). Whole animal metabolic health assessment by **I** GTT and **J** ITT evaluations (*n* = 5/group). Data are shown as bar graphs with SEM of five mice per control or treated group and are representative of two independent experiments. For body weight, GTT (group: *P* = 0.4298, time: *P* = < 0.0001 and group × time interaction: *P* = < 0.0889) and ITT (group: *P* = 0.0004, time: *P* = < 0.0001 and group × time interaction: *P* = < 0.0534) statistical significance was determined by repeated measure Two-way ANOVA post-hoc. For bar graphs, data normality status was used to determine statistical analysis. Normal data statistical significance was determined by parametric student's unpaired *t* test, while not normally distributed data were analyzed using non-parametric Mann-Whitney test. ns = *P* > 0.05, *=*P* < 0.05, ** =*P* < 0.01, ***P* < 0.001.

the modulation of systemic inflammation and tissue growth. In line with this, increased hepatic expression of *Tgm2* post-infection has been implicated in development of liver fibrosis[52]; thus, further studies investigating TGM2 as a potential candidate regulating liver steatosis/fibrosis status in obesity, would contribute to delineating the holistic role of TGM2 in diet-induced obesity.

Interestingly, our evaluation of monocytes found increased *Tgm2* intracellular expression in HFD eWAT monocytes, compared to CD controls. Literature reports have previously shown the direct role of TGM2 in regulating monocyte adhesion and extravasation properties during high inflammatory settings[16]. Importantly, increased TGM2 expression in monocytes leads to enhanced macrophage differentiation[18,53], which would

serve as a strong indication that TGM2+ATMs may result from an increased monocyte-derived infiltrating population with elevated *Tgm2* expression. More detailed studies on changes in the AT monocyte population at different stages of the macrophage differentiation process would help test this hypothesis.

TGM2 is a well-established marker of anti-inflammatory M2 macrophages[19]. Previous works have reported increased abundance of TGM2+ M2 macrophages in highly pro-inflammatory disease states, such as asthma, type 1 diabetes, sepsis, and celiac disease[19,54,55]. Concomitantly, other reports have demonstrated that M2 CD206+ ATMs have a greater prevalence in human diabetes[56] and help regulate glucose metabolism[45]. This goes in line with our observation of increased anti-inflammatory

TGM2+ ATMs in diet-induced obese mice with high levels of pro-inflammation. It appears, however, that the TGM2 macrophage anti-inflammatory effects are insufficient to reverse the commonly observed pro-inflammatory responses of the HFD-induced obese mouse model, perhaps due to TGM2 effects being suppressed by sustained chronic pro-inflammation. Emerging reports propose a new paradigm suggesting that pro- and anti-inflammatory responses could be simultaneously present during early stages of induced inflammation with the goal to prevent hyperinflammation (or cytokine storm). In turn, presence of both anti- and pro- inflammatory systems would eventually result in excessive simultaneous pro- inflammation and immune suppression events, resulting in a state of low-grade chronic inflammation[57].

Our analysis investigating the transcriptional profile of *Tgm2*+ ATMs identified *Ctsd*, *Ctsl*, and *Mmp12* genes to be significantly upregulated in the HFD *Tgm2*+ ATMs. Importantly, macrophage-derived *Ctsd* has recently been identified as a key suppressor of liver fibrosis via modulation of collagen remodeling and immune responses in vivo[58]. Similarly, *Ctsl* expression is elevated in the lipid-associated macrophage (LAM) population[59], which is well-characterized by its anti-inflammatory effects in regulating phagocytosis and endocytosis leading to the idea that the LAM populations possesses a protective role in obesity-induced unhealthy AT[60,61]. In the case of *Mmp12*, an important regulator of extracellular matrix and wound healing responses that degrades basement membrane laminin[62], a previous ablation study showed its role in modulating endothelial cell dysfunction via increased extracellular matrix accumulation during tissue fibrosis stage. Conversely, *Scl9a9* gene expression was significantly downregulated in the *Tgm2*+ ATMs subset, indicating the potential for this ATM subset to possess pro-inflammatory activities, at least in the context of bacterial killing, given Sclc9a9's role in modulating phagosome maturation via altering of the lumenal pH[63]. Altogether, *Tgm2* expression in ATMs seems to provide pro-resolving macrophage properties involved in increased accumulation of tissue extracellular matrix, which is commonly linked to increased tissue fibrosis status that results in IR and metabolic dysfunction[64].

Under M0/M1/M2 polarizing conditions, *Tgm2*-silenced BMMs had decreased expression of CD206, which goes in line with previous reports showing reduced CD206 expression via PPARγ (a known promoter of M2 alternatively activated macrophages) in *Tgm2*-silenced macrophages refs. [65,66]. Furthermore, our findings showing decreased IL-10, and increased MHC Class II and CD64, corroborate previous works showing that TGM2 inhibition in macrophages results in increased pro-inflammatory responses with perturbed IL-10 secretion and reduced efferocytosis capacities[25]. Future studies on *Tgm2* silenced metabolically activated, oxidized and/or lipid-associated macrophages are required to test for reproduced findings in M1/M2 polarized BMMs.

We found *Tgm2* silencing led to a significant increase in the CD206-CD11c+ population along with a significant decrease in the CD206+ CD11c- population, a recently identified plastic macrophage subset associated with IR in humans[56,67], suggesting that release of TGM2 by ATMs contributes to the modulation of inflammatory ATMs plasticity in obesity. This is strongly believed, as previous works have shown that although CD11c marker expression remains constant along resolution of inflammation, pro-inflammatory responses may concomitantly begin to adopt expression of anti-inflammatory markers[13,68] along with CD11c. Future studies employing inducible conditional *Tgm2* silencing mouse models would supply information about its effects on appearance of CD206+ CD11c+ ATMs along the course of obesity. Furthermore, our co-culture studies showed a significant decrease in the co-cultured activated AT IL-10+ CD4+ T cells in the presence of *Tgm2* silenced BMMs, suggesting the hypothesis that macrophage-derived TGM2 regulation of IL-10 secretion by CD4+ T cells may also indirectly regulate CD206 expression in neighboring macrophages, via IL-10R activation. Subsequent studies should employ conditional IL-10 knockout mice to investigate the TGM2/IL-10 axis in modulating CD206 expression in ATMs.

We also aimed to determine if extracellular TGM2 in absence of other macrophage-derived factors would be one of the key modulators of AT leukocyte inflammatory profile. We observed increased CD4+ IL-10+ T cells only in the unstimulated group, while this effect was not observed in the anti-CD3/CD28 activated T cells, potentially suggesting that the increased IL-10 secretion in these groups pertains to TGM2-induced IL-10 secretion during the 24 h incubation period prior to the addition of activation stimuli. Follow-up work investigating effects of macrophage secreted TGM2 on self and surrounding ATMs, namely via abolishment of TGM2 release in vitro using a blocking antibody, would be needed to fully establish the TGM2 immunometabolic role in regulating AT immune cell inflammation. Furthermore, treatment of AT leukocytes from HFD mice with rTGM2 did not show significant changes in IFNγ or IL-10 expressing T cells under any activation stimuli, which makes it likely that the effects of TGM2 in established obesity are dampened by other longstanding obesity-related factors.

Lastly, our in vivo mouse model silencing *Tgm2* in CD11b+ myeloid cells employed the widely used lentiviral vector approach to manipulate gene expression in vivo; our data showed that about 50% of the myeloid AT cell population was successfully silenced for *Tgm2*. Although this silencing efficiency was sufficient to yield a detectable phenotype, the intraperitoneal injection of lentivirus particles most likely allowed for a wider spread/distribution into tissues other than the AT, thereby reducing its potential for a greater silencing effect within the AT myeloid cell population. Surgical approaches employing directly injected lentivirus particles into the target fat pad could increase silencing efficiency within the AT site[69]. However, intraperitoneal injections of lentivirus particles allowed targeting of both tissue resident and infiltrating myeloid cell populations, as they dynamically change along the course of HFD, thereby providing a more ample view of *Tgm2* silencing effects in myeloid cells during obesity. Future studies employing a transgenic mouse model with germline silencing of *Tgm2* in the myeloid cell populations could ensure complete silencing efficacy.

Our in vivo results upon reduction of *Tgm2* in macrophages corroborate our in vitro data showing an increased pro-inflammatory AT profile, given increased total leukocyte infiltration and ATMs expressing high levels of MHC class II markers, while no significant effects were observed in the IL-10+ATMs nor in the ATT populations, potentially due to previously reported effects of HFD treatment on ATT exhaustion[46]. AT inflammation contribution to AT and systemic IR[70] is known to occur via paracrine effects of inflammatory cell-derived factors on insulin signaling and metabolism in adipocytes. We hypothesize that an increase in pro-inflammatory ATMs[71] and leukocyte AT infiltration, along with elevated levels of systemic MCP-1[72], TNFα, and reduced IL-27, are key elicitors of IR and increased body weight, fat mass, and systemic adiposity in mice silenced for TGM2 in myeloid cells

Altogether, this work contributes to the field's knowledge of TGM2 as a marker identifying a subset of ATMs acting as potential inert counter regulators of inflammation, that is otherwise responsible for development of obesity and IR.

## Data availability

The datasets generated/analyzed for this study can be found in the Mendeley Data repository: Yanovski, Jack; Elizondo, Diana (2023), "Elizondo et al. TGM2-expressing macrophages modulate AT inflammation", Mendeley Data, V1, https://doi.org/10.17632/5t8kdsjsw5.1. Uncropped/unedited gels for respective figures can be found in supplementary information.

## Code availability

Access to processed dataset generated by Sarvari et al., 2021 is available in the Open Science framework: tsjqc and the single cell portal: SCP1179.

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

## Acknowledgements

The authors thank Dr. K.O. for providing continuous feedback on this work, and the Flow Cytometry Core of the National Heart, Lung, and Blood Institute for flow cytometry data acquisition and the NICHD's Bioinformatics and Scientific Programming Core. This work has been supported by the Intramural Research Program of the Eunice Kennedy Shriver National Institute of Child Health and Human Development (ZIAHD00641, to J.A.Y.) with supplemental funding from the Maximizing Opportunities for Scientific and Academic Independent Careers Postdoctoral Career Transition to Promote Diversity (1K99DK136921) and NICHD Early Career Investigator Awards (to D.M.E.).

## Author contributions

D.M.E. designed studies, researched data, and wrote the manuscript. T.P.P.: research data. B.C.: research data. E.J.: research data. J.C.: research data. M.C.H. researched data. A.M. analyzed data. J.A.Y. designed studies and wrote the manuscript. All authors participated in editing the manuscript and approved the final draft. D.M.E. and J.A.Y. are the guarantors of this work and, as such, had full access to all the data in the study and take responsibility for the integrity of the data and the accuracy of the data analysis.

## Competing interests

The authors declare no competing interests.
