## [Transparent peer review file · Communications Biology]

Transglutaminase 2-expressing macrophages modulate adipose tissue inflammation.

Corresponding Author: Dr Jack Yanovski

Version 0:

Reviewer comments:

Reviewer #1

(Remarks to the Author)

The article by Elizondo et al presents a potentially interesting finding that TGM2-expressing macrophages could modulate adipose tissue inflammation. The authors found that macrophage-derived TGM2 served as an anti-inflammatory driver, failing to counteract chronic pro-inflammation in obesity adipose tissue. However there are major defects in this manuscript. Some concerns regarding data interpretation and results require additional experiments and figures to demonstrate the concepts articulated in the manuscript.

1. Figure 1 should be moved to supplementary materials as it is only a simple demonstration of previously known results.
2. Figure 2, the reason for studying the differential expression of glutamine transaminase and TGM2 is insufficient. More evidences such as proteomics should be provided.
3. Figure 2D, mice fed with chow diet should be used as negative controls.
4. Figure S2, the author should clarify why there is no significant difference in the expression of TGM2 in BMMs between CD and HFD mice.
5. Figure 3, the purpose of stimulation of anti-CD3/CD28 activation beads, or the relationship between anti-CD3/CD28 activation and SVF cells should be clarified.
6. Throughout the paper, the authors should use scale bars rather than magnification in the analysis of fluorescence microscopy data.
7. The regulation of inflammation spectrum by TGM2 has only been confirmed in vitro. Please provide more in vivo evidences using knockout mice.
8. The existing data is insufficient to support the conclusion that TGM2 is the sole macrophage-derived key intermediary inducer of anti-inflammatory properties. Please provide more sufficient or necessary evidences.

Reviewer #2

(Remarks to the Author)

The paper by Elizondo et al., reveals a role of transglutaminase 2 (TGM2) in adipose tissue macrophages (ATMs). The here presented data provide solid evidence that TGM2 is expressed in ATMs. TGM2 expression and activity in the whole adipose tissue increase in obesity. TGM2 silencing with CRISPR-Cas9 technology in bone marrow-derived macrophages (BMDMs) increases IFN γ secretion and MHC-II expression in steady state and decreases CD206 expression upon IL4 treatment. Moreover, TGM2 silencing in BMDMs reduced the abundance of CD206+ stromal vascular fraction (SVF)-associated ATMs in a co-culture system of BMDMs with SVF cells. TGM2 silencing in BMDMs also reduced IL10 expression in SVF associated-T cells in the same co-culture system. Application of recombinant TGM2 to the co-culture system increased IL10 production in T cells at steady state and upon anti-CD3/CD28 activation.

Major comment:

Although these findings are interesting and convincing in terms of TGM2 expression in ATMs, in its current state the study is quite preliminary since conclusions on the role of TGM2 in the adipose tissue are exclusively based on in vitro data. Instead, only in vivo data using myeloid-specific TGM2 deficient mice with diet-induced obesity would provide solid proof for a role of TGM2 in adipose tissue inflammation and metabolic disease. The *Tgm2^{fl/fl}* mouse line exists and could be crossed with the *LysM-Cre* mouse line. Assessment of adipose tissue pathophysiology (inflammation, fibrosis, hypertrophy) and organismal metabolism (insulin tolerance, energy consumption) in *LysM-Cre;Tgm2^{fl/fl}* mice fed with a control or a high-fat diet (CD,

HFD) would reveal the 'true' role of ATM-derived TGM2.

It is mentioned that TGM2 could be expressed in other SVF cell populations, however this was not tested. Examination of TGM2 expression in preadipocytes, endothelial cells and immune cells other than ATMs would be an important addition to the paper.

Furthermore, is the increase of TGM2 with obesity unique for the adipose tissue or does it also occur in other metabolic organs and tissues (liver, muscle)?

Transcriptomic analysis of TGM2+ and TGM2- ATMs could provide insight in the TGM2-dependent ATM plasticity.

Figure 1 should be in the supplement

Minor comments:

The manuscript requires thorough editing, some examples:

Line 26: 'stromal vascular fraction (SVF) undergoing cell activation.': unclear what is meant by 'cell activation'

Line 37: 'with inflammatory-driving immune cells': 'pro-inflammatory immune cells'

Lines 186-7: 'IL-10...CD45, TCR β , CD4, 187 CD25, IL-10,'

Lines 337-339: The designed studies and presented data in this paragraph, do not answer the question whether TGM2 is the only macrophage-derived key intermediary inducer of anti-inflammatory properties.

The possible mechanisms mediating the effects of TGM2 could be discussed in the Discussion section based on reports in other cell types.

Version 1:

Reviewer comments:

Reviewer #1

(Remarks to the Author)

The authors have addressed all of my concerns

Reviewer #2

(Remarks to the Author)

Version 2:

Reviewer comments:

Reviewer #2

(Remarks to the Author)

The authors have convincingly addressed all my comments.

Reviewer #1:

The article by Elizondo et al presents a potentially interesting finding that TGM2-expressing macrophages could modulate adipose tissue inflammation. The authors found that macrophage-derived TGM2 served as an anti-inflammatory driver, failing to counteract chronic pro-inflammation in obesity adipose tissue. However there are major defects in this manuscript. Some concerns regarding data interpretation and results require additional experiments and figures to demonstrate the concepts articulated in the manuscript.

COMMENT #1. Figure 1 should be moved to supplementary materials as it is only a simple demonstration of previously known results.

Response: We thank the reviewer for this suggestion and have moved original Figure 1 to Supplemental Figure S1. Manuscript text has been revised, accordingly.

COMMENT #2. Figure 2, the reason for studying the differential expression of glutamine transaminase and TGM2 is insufficient. More evidences such as proteomics should be provided.

Response: We understand this concern, and we agree that an unbiased proteomic approach identifying TGM2 increased expression in eWAT HFD would have been adequate to begin studying TGM2 in obesity. Unfortunately, we have not been able to find such data through the literature (i.e. eWAT proteomics from CD vs. HFD male mice), nor it has been generated in our hands. Given our interest in finding candidate modulators of obesity-induced AT dysfunction, we began our focus on transglutaminases F13A1 and TGM2, due to appearance of literature reports showing their role in regulating adipogenesis [PMID: 26313919, 24934257, 27759118] with initial attention to TGM2, as shown in this work. Moreover, navigation of the Human protein atlas showed expression of TGM2 in adipose tissue macrophages (please see below image), a population of high interest in our research focus, which lead us to first ask the question whether TGM2 expression pattern changes in HFD. Nonetheless, we have now included a secondary analysis of published snRNAseq data on the eWAT from CD vs. HFD mice and included these results in the new Fig 1D-1E in hopes of making our choice of TGM2 study less arbitrary.

COMMENT #3. Figure 2D, mice fed with chow diet should be used as negative controls.

Response: Thank you for suggesting this important control. We have replaced Figure 2D with new data including qPCR analysis from CD vs. HFD sorted eWAT subpopulations (preadipocytes, macrophages and monocytes) now as Figure 1I. Additionally, we have incorporated flow cytometric data evaluating TGM2 expression in stromal vascular fraction macrophages from CD vs. HFD mice as Figure G-H. Importantly, we have expanded on this analysis by including a secondary analysis of CD vs. HFD snRNAseq data by Sarvari, et., al. 2021 (NCBI GEO: GSE160729) to evaluate Tgm2 expression in CD vs. HFD conditions for macrophages, dendritic cells, B cells and T cells. Data have been added as Figures 1D-1E. We believe that these new data provide a more accurate data interpretation of Tgm2 expression in CD vs. HFD various adipose tissue populations. Manuscript text has been revised, as follows:

Introduction:

[Lines: 62-64] *“Transglutaminase 2 (TGM2) is a protein ubiquitously expressed in multiple cell types, including monocytes, M2 macrophages^{14, 15, 16, 17, 18, 19}, thymocytes²⁰, myoblasts²¹, endothelial cells²², and preadipocytes²³ among others.”*

Methods:

[Lines: 231-243] *“SVF was harvested from eWAT, as described³⁵ and utilized for: 1) staining of F4/80+TGM2 ATMs, 2) co-culture studies or 3) AT cell sorting. For staining approach of SVF, co-cultured or rTGM2-treated SVF, cells were harvested and incubated in Fc block Anti-Mouse CD16/CD32 (BD Biosciences; Cat#553141) 1:50 for 5 min on ice and washed 1x in FACS. Next, cells were incubated in primary conjugated antibodies diluted to 1:100 in FACS, unless otherwise specified for 20min at 4°C, as follows: MHC Class II-PerCp-eFluor710 (Thermo Fisher; Cat#46-5321-82), F4/80-PE Cy.7 (Biolegend; Cat#123113), CD206-Brilliant Violet 421 (Biolegend; Cat#141717), CD11c-Alexa Fluor 594 (Biolegend; Cat#117346), TCRβ-PE Cy.7 (Biolegend; Cat#109222), CD45- Alexa Fluor 660 (Thermo Fisher; Cat#606-0451-82), CD4-Alexa 488 (Biolegend; Cat#100425), CD25-PE Cy.5 (Biolegend; Cat#102010), IFNγ-PE/Dazzle (Biolegend; Cat#505846), IL-10- APC Cy.7 (Biolegend; Cat#5050335), IL-10- BV421 (Biolegend; Cat#505022), TCRβ- PercP Cy5.5 (Biolegend; Cat#109227), IFNγ-BV421 (Biolegend; Cat#505022).”*

[Lines: 223-228] *“2.13 eWAT snRNAseq secondary analysis: We downloaded processed Seurat objects from Single Cell Portal accession ID SCP1179 generated by Sarvari et al 2021³⁴ containing data from 8355 immune cells (‘eWAT_Immune’). 309 out of 8355 cells (3.69%) expressing Tgm2 were labeled as Tgm2+. Cells with zero Tgm2 expression were labeled as Tgm2-. We then performed differential expression comparing Tgm2+ cells to Tgm2- cells using the Wilcoxon rank-sum test via the FindMarkers function of the Seurat R package v4.3.0.”*

Results:

[Lines: 323-328] *“To determine the source of AT TGM2 in HFD-induced obesity, a secondary analysis of snRNAseq eWAT published data by Sarvari et.al.³⁴, showed an increased abundance of Tgm2 expressing cells belonging to the immune cell fraction in HFD mice compared to CD group (Fig. 1D). Importantly, when narrowing down the analysis to focus on the immune cell populations, ATMs were the subset that showed a higher number of cells positive for Tgm2 in the HFD group compared to CD control (Fig. 1E).”*

[Lines: 329-334] *“Corroborative flow cytometric assessment in SVF from CD or HFD mice (gating strategy shown in Supplementary Fig. S2B) showed an increased abundance of F4/80+ ATMs (Fig. 1G), that co-expressed TGM2 (Fig. 1H) (as tested with a specific TGM2 antibody shown in Supplementary Fig. S2C-S2E). There was an increased number of TGM2+ATMs in the eWAT from HFD mice compared to CD controls.”*

[Lines: 336-343] *“Furthermore, to investigate if intracellular levels of Tgm2 differed in CD vs. HFD eWAT-derived cell populations, eWAT-derived SVF cells were sorted for preadipocytes, macrophages and monocytes (gating strategy in Supplementary Fig. S2F). No significant differences in intracellular Tgm2 expression by qPCR analysis were observed in CD vs. HFD preadipocytes or macrophages, while a significant increase in*

intracellular Tgm2 was shown in HFD eWAT-SVF sorted monocytes, compared to CD controls (Fig. 1I). This indicates that most of the increased TGM2 in the eWAT from HFD male mice emanates from an increased number of TGM2+ATMs and the presence of monocytes expressing high intracellular levels of Tgm2 compared to CD controls.”

Discussion:

[Lines: 578-585] *“Although our studies focused mostly on TGM2 expression in eWAT-derived SVF leukocytes, our secondary analysis of eWAT snRNAseq data showed Tgm2 expression in FAP, adipocyte, and endothelial cell populations from both CD and HFD eWAT, indicating that ATMs are not the sole contributor of TGM2 in AT during HFD conditions. However, it was the Tgm2+ATMs population that showed an obvious increase in cell numbers in the HFD group compared to CD control, suggesting that the increased macrophage population in eWAT from HFD serves as an extra source of TGM2 that significantly augments the overall expression of TGM2 at the tissue level.*

[Lines: 591-593] *“Further studies modulating TGM2 expression in other eWAT cell populations, or in the liver would greatly contribute to delineating the holistic role of TGM2 in diet-induced obesity metabolic dysfunction.”*

[Lines: 608-618] *“However, our data evaluating Tgm2 intracellular expression levels in monocytes also showed increased Tgm2 expression in HFD eWAT monocytes, compared to CD controls. Literature reports have previously shown the direct role of TGM2 in regulating monocyte adhesion and extravasation properties during high inflammatory settings¹⁶. Importantly, increased levels of TGM2 expression in monocytes leads to enhanced macrophage differentiation^{18, 50}, which would serve as a strong indication that TGM2+ATMs may result from an increased monocyte-derived infiltrating population that begins to have increasing Tgm2 expression along with macrophage differentiation/maturation/activation while residing in AT. A more detailed study ascertaining changes in the AT infiltrating monocyte population at different stages of the macrophage differentiation process would help test this hypothesis.”*

COMMENT #4. Figure S2, the author should clarify why there is no significant difference in the expression of TGM2 in BMMs between CD and HFD mice.

Response: We apologize for the lack of clarity in conveying this data interpretation. To better address this finding, we have added new data (based on comment 3) showing no significant differences in intracellular levels of Tgm2 in ATMs from CD vs. HFD eWAT. Additionally, inclusion of our new secondary analysis of eWAT snRNAseq published data further corroborates that there are similar Tgm2 expression levels in eWAT ATMs from CD vs. HFD, while the number of individual cells is notably increased in the HFD group. which resembles our previous observation in the BMMs. We believe that the increased tissue expression of TGM2 corresponds to elevated numbers of TGM2+ATMs, as opposed to increased intracellular expression of the gene. We have updated manuscript text to reflect incorporation of supporting data and to better explain this data interpretation, as follows, (and as shown in comment 3 response for methods and results revised sections)

Results:

[Lines: 375-381] *“We used BMMs from CD mice because no significant differences in TGM2 expression in differentiated BMMs from CD vs. HFD mice were observed by fluorescence microscopy (Supplementary Fig. S3A), flow cytometric or qPCR analysis (Supplementary Fig. S3B-S2F), confirming previously discussed data showing that although Tgm2 intracellular levels remain equal in macrophages from CD vs. HFD treatment groups, observed differences in total tissue TGM2 may be due to the increased number of TGM2+ macrophages.”*

COMMENT #5. Figure 3, the purpose of stimulation of anti-CD3/CD28 activation beads, or the relationship between anti-CD3/CD28 activation and SVF cells should be clarified.

Response: We apologize for the lack of clarity on this concept – one of the goals of this work was to evaluate macrophage-derived TGM2 as a potential secreted molecule mediating adipose tissue leukocyte-macrophage cross talk in AT. Therefore, to ensure secretion of TGM2 in our *in vitro* experiments, we needed to recapitulate

observed HFD pro-inflammatory responses that we commonly see in HFD AT. To do so, we aimed to induce inflammatory settings by activating AT T cells with stimulation of anti-CD3/CD28 activation beads, which would then mount pro-inflammation. Next, we concomitantly interrogated if presence or absence of secreted TGM2 impaired AT T cell activation responses. We have updated manuscript text, as follows:

Results:

[Lines: 430-435] “. To stimulate Ca^{2+} - dependent TGM2 secretion/activity by co-cultured BMMs³⁹, we induced a pro-inflammatory environment in the shared supernatant between both cell cultures in the transwell plate. To do so, we increased intracellular Ca^{2+} levels within SVF co-cultured AT T cells to mimic pro-inflammation seen in HFD AT, via stimulation with anti-CD3/CD28 activation beads for 72h⁴⁰.”

COMMENT #6. Throughout the paper, the authors should use scale bars rather than magnification in the analysis of fluorescence microscopy data.

Response: Thank you – we have added scale bars to all microscopy analysis in the paper (Figure 1F, Figure 6G and Supplemental Fig S3A).

COMMENT #7. The regulation of inflammation spectrum by TGM2 has only been confirmed in vitro. Please provide more in vivo evidence using knockout mice. & COMMENT #8. The existing data is insufficient to support the conclusion that TGM2 is the sole macrophage-derived key intermediary inducer of anti-inflammatory properties. Please provide more sufficient or necessary evidence.

Response: We agree with reviewer, and we have now added new in vivo data supporting our in vitro observations – please see detailed response below to Reviewer 2 Comment #1.

Reviewer #2:

The paper by Elizondo et al., reveals a role of transglutaminase 2 (TGM2) in adipose tissue macrophages (ATMs). The here presented data provide solid evidence that TGM2 is expressed in ATMs. TGM2 expression and activity in the whole adipose tissue increase in obesity. TGM2 silencing with CRISPR-Cas9 technology in bone marrow-derived macrophages (BMDMs) increases IFN γ secretion and MHC-II expression in steady state and decreases CD206 expression upon IL4 treatment. Moreover, TGM2 silencing in BMDMs reduced the abundance of CD206+ stromal vascular fraction (SVF)-associated ATMs in a co-culture system of BMDMs with SVF cells. TGM2 silencing in BMDMs also reduced IL10 expression in SVF associated-T cells in the same co-culture system. Application of recombinant TGM2 to the co-culture system increased IL10 production in T cells at steady state and upon anti-CD3/CD28 activation.

Major comment:

Although these findings are interesting and convincing in terms of TGM2 expression in ATMs, in its current state the study is quite preliminary since conclusions on the role of TGM2 in the adipose tissue are exclusively based on in vitro data.

COMMENT #1) Instead, only in vivo data using myeloid-specific TGM2 deficient mice with diet-induced obesity would provide solid proof for a role of TGM2 in adipose tissue inflammation and metabolic disease. The Tgm2fl/fl mouse line exists and could be crossed with the LysM-Cre mouse line. Assessment of adipose tissue pathophysiology (inflammation, fibrosis, hypertrophy) and organismal metabolism (insulin tolerance, energy consumption) in LysM-Cre;Tgm2fl/fl mice fed with a control or a high-fat diet (CD, HFD) would reveal the ‘true’ role of ATM-derived TGM2.

Response: We concur with reviewer that addition of in vivo analyses would add considerable data for our model. We are currently in the process of generating a LysM-Cre x Tgm2fl/fl mouse model. However, given time requirements to generate the desired genotypes, and the fact that there is expression of LysM gene in cells other than macrophages, such as neurons [PMID: 27062494; 36115548], which could drastically impact our phenotype if neuronal expression has consequences for eating behavior and/or energy expenditure, we opted to employ a lentivirus-mediated approach via intraperitoneal injection of lentiviruses carrying a myeloid-specific CD11b promoter upstream of Cre recombinase with EGFP reporter, or lentivirus control, containing human EF1A promoter upstream of mCherry, followed by CMV promoter upstream of EGFP. These lentivirus particles were injected into Tgm2fl/fl homozygous mice with the premise of inducing Cre recombinase, thereby leading to silencing of Tgm2 within CD11b myeloid cell populations. Data have been added as new Fig 5, Fig 6 and Supplemental figure S6. Text in manuscript has been revised, as follows:

Abstract:

[Lines: 32-36] *“In vivo silencing of TGM2 in CD11b+ cells in HFD mice resulted in augmented pro-inflammation both in eWAT and serum, and increased adiposity and insulin resistance. This suggests that macrophage-derived TGM2 serves as an anti-inflammatory driver that fails to completely counteract chronic pro-inflammation in obesity.”*

Introduction:

[Lines: 83-91] *“Lastly, an in vivo mouse model silencing TGM2 in CD11b myeloid cells in HFD mice further demonstrated that reducing TGM2 in myeloid cells results in increased pro-inflammatory responses, along with increased obesity and insulin resistance (IR).”*

“Taken together, our results demonstrate that TGM2 expression in macrophages exerts anti-inflammatory properties in AT lymphocytes, thereby protecting against the development of insulin resistance. These data highlight the role of TGM2-producing macrophages in balancing the AT inflammation and metabolic dysfunction induced by HFD obesity.”

Methods:

[Lines: 115-126] “2.3 *In vivo* TGM2 silencing in CD11b myeloid cells: Lentivirus carrying CD11b upstream of Cre-recombinase with EGFP reporter downstream of CMV promoter pLV[Exp]-EGFP-CD11b>Cre [Cat:VB230425-1457ejv] (pLV.CD11b-Cre), or respective lentivirus control carrying human EF1A promoter upstream of an mCherry reporter to detect potential non-specific gene induction, followed by CMV promoter upstream of EGFP reporter pLV[Exp]-EGFP/Puro-EF1A>mCherry [Vector ID: VB010000-9298rtf] (pLV.Control) were purchased from Vector Builder (Chicago, IL). EGFP was utilized to evaluate transduction efficiency. Lentiviruses were purchased ultra-purified for *in vivo* applications, diluted in sterile PBS to a final concentration of 10^8 TU/mL in 250uL. Lentiviruses were injected intraperitoneally in Tgm2 floxed homozygous mice that were 5 weeks old. Mice were placed on a 60%HFD one-week post-pLV injections and monitored for changes in weight, glucose and insulin responses by glucose or insulin tolerance tests, along with changes in whole body composition by DEXA scanning.”

[Lines: 135-137] “Digital X-ray system Ultra focus DXA Faxitron (Tucson, AZ) was used for evaluation of whole-body composition in mice post-lentivirus and diet-treatments.”

[Line: 273] 2.15 ELISA/LEGENDplex:

[Lines: 276-278] “Multiparametric LEGENDplex kit was used to detect IFN γ , TNF α and IL-6, MCP-1, IL-10, IL-17A, IFN γ , TNF α , IL-23, IL-27 or IFN β in both harvested supernatant and mouse serum (Biolegend; 740150).”

Results:

[Lines: 511-564] “3.5 *In vivo* CD11b myeloid cell silencing of TGM2 results in increased tissue and systemic inflammation, accompanied by increased diet-induced obesity and insulin resistance. To continue investigating the effects of TGM2 expression in myeloid cells on diet-induced obesity inflammation and metabolic dysfunction, we utilized a lentivirus approach to induce silencing of TGM2 in CD11b+ myeloid cells. Briefly, we employed a lentivirus control (pLV.Control) carrying human EF1A promoter upstream of mCherry and EGFP reporter downstream of ubiquitously expressed CMV promoter, as well as a lentivirus to silence TGM2 in CD11b+ cells (pLV.CD11b-Cre) via expression of CD11b promoter upstream of Cre recombinase and EGFP reporter downstream of CMV promoter (**Fig. 5A**). Lentiviruses were intraperitoneally injected into Tgm2 floxed homozygous mice at 5 weeks of age and placed on a 60% HFD one-week post-injections and monitored for body weight changes on a weekly basis. Additional assessments included GTT and ITT, whole body composition by DEXA scan and flow cytometric analysis of AT SVF immune cell profile, as denoted in the study timeline (**Fig. 5B**).

We corroborated that silencing of TGM2 in CD11b+ myeloid cells within the AT resulted in AT immune cell profile changes via flow cytometric assessment in the SVF, as shown in gating strategy (**Fig. 5C**). There was a significant reduction in TGM2+ATMs (CD45+CD11b+GFP+F4/80+MHC II+TGM2+), a significant increase in total leukocytes (CD45+) and in MHC IIhi+ATMs (CD45+CD11b+GFP+F4/80+MHC IIhi+) in the pLV.CD11b-cre injected mice, compared to pLV.Control, while no significant differences were observed in the MHCIIlo+ATMs (CD45+CD11b+GFP+F4/80+MHC IIlo+), IL-10+ATMs (CD45+CD11b+GFP+F4/80+MHC II+IL-10+), or AT T cells (CD45+CD11b-GFP-F4/80-MHC IIhi-TCRb+) between groups (**Fig. 5D**). Our data show that about 60% of the macrophages were successfully silenced for TGM2 upon lentivirus incorporation and that this reduction of TGM2+ expressing ATMs leads to an increase in pro-inflammatory cell profile within the AT of diet-induced obese mice. Next, assessment of serum from pLV.Control vs. pLV.CD11b-Cre injected mice showed an increased inflammatory cytokine profile, as shown by significantly higher levels of MCP-1 and TNF α , while no significant differences were observed in IL-10, IL-17A, IFN γ , IL-23 or IFN β between the groups (**Fig. 5E**). Interestingly, data showed a significant downregulation of circulating IL-27 in the pLV.CD11b-Cre injected mice compared to pLV.Control, which despite possessing both pro- and anti-inflammatory properties⁴⁴, has been proven to positively modulate weight gain, adiposity and insulin resistance⁴⁵. Altogether, these data suggest that silencing of TGM2 in CD11b+ myeloid AT cells result in increased inflammation in diet-induced obese male mice.

Concomitantly, to evaluate the metabolic health effects of TGM2 silencing in CD11b myeloid cells for both male and female mice, we evaluated body weight changes on a weekly basis, finding a significant increase in whole

body weight from male mice injected with pLV.CD11b-Cre compared to pLV.Control mice (Fig. 6A). Furthermore, when evaluating whole body composition by DEXA scanning, results showed no significant changes in lean mass, while fat mass was shown to be increased in the pLV.CD11b-Cre male (Fig. 6C-D) and female (Supplementary Figure S6D-E) mice. To evaluate if changes in eWAT mass were contributing to this increased fat mass observed by DEXA, we assessed gross eWAT and liver tissue weight post-mortem, where results showed increased eWAT tissue weight (Fig. 6E; Supplementary Figure S6F) in the pLV.CD11b-Cre male and female mice compared to pLV.Control, while no significant weight differences were apparent in the liver (Fig. 6F; Supplementary Figure S6G). Further histological assessment of eWAT stained for H&E (Fig. 6G) showed a significant reduction in adipocyte number and increased adipocyte size in the pLV.CD11b-Cre compared to pLV.Control male mice (Fig. 6H). We saw no significant differences in glucose sensitivity (Fig. 6I) in pLV.CD11b-Cre compared to pLV.Control male mice, while insulin sensitivity was significantly impaired in pLV.CD11b-Cre mice compared to pLV.Control (Fig. 6J). Female metabolic assessments mirrored findings in male data (Supp Fig. S6J-K), indicating that TGM2 silencing in CD11b+ myeloid cells play a significant role in the pathophysiology of obesity, suggesting an absence of important sexual dimorphisms. Altogether, these data indicate that TGM2 expression in ATMs is an essential player modulating AT inflammation associated with metabolic dysfunction.”

Discussion:

[Lines: 762-792] “Lastly, our *in vivo* mouse model silencing TGM2 in CD11b+ myeloid cells employed the widely used lentiviral vector approach to manipulate gene expression *in vivo*. Percentage of lentivirus transduction/silencing efficiency is a fundamental determinant of observed outcomes. Our data showed that about 50% of the myeloid AT cell population was successfully silenced for TGM2. Although this silencing efficiency was sufficient to yield a detectable phenotype, the intraperitoneal injection of lentivirus particles most likely allowed for a wider spread/distribution into tissues other than the AT, thereby reducing its potential for a greater silencing effect within the AT myeloid cell population. More invasive surgical approaches employing directly injected lentivirus particles into the target fat pad could increase silencing efficiency within the AT site⁶⁹. However, intraperitoneal injections of lentivirus particles allowed targeting of both tissue resident and infiltrating myeloid cell populations, as they dynamically change along the course of HFD, thereby providing a more ample view of TGM2 silencing effects in myeloid cells during obesity. Future studies employing a transgenic mouse model with germline silencing of TGM2 in the myeloid cell populations could ensure complete silencing efficacy.

Our *in vivo* results upon inhibition of TGM2 in macrophages corroborate our *in vitro* data showing an increased pro-inflammatory AT profile, given increased total leukocyte infiltration and ATMs expressing high levels of MHC class II markers, while no significant effects were observed in the IL-10+ATMs nor in the AT T cell populations, potentially due to previously reported effects of HFD treatment on AT T cell exhaustion⁴³.

AT inflammation is well-established to contribute to AT and systemic insulin resistance⁷⁰. This is known to occur via paracrine effects of inflammatory cell-derived factors on insulin signaling and metabolism in adipocytes. We hypothesize that our observed increase in pro-inflammatory ATMs⁷¹ and leukocyte AT infiltration, along with elevated levels of systemic MCP-1⁷², TNF α ³⁶, and reduced IL-27, are key elicitors of insulin resistance and increased body weight, fat mass and systemic adiposity in mice silenced for TGM2 in myeloid cells. Altogether, these data support the hypothesis that there is a role of macrophage-derived TGM2 in regulating both AT and systemic inflammation responsible for causing metabolic dysfunction in obesity.”

Scope statement:

[Line: 833] “...along with a novel *in vivo* lentivirus-mediated approach to silence TGM2 in myeloid cells...”

COMMENT #2) It is mentioned that TGM2 could be expressed in other SVF cell populations, however this was not tested. Examination of TGM2 expression in preadipocytes, endothelial cells and immune cells other than ATMs would be an important addition to the paper.

Response: We thank the Reviewer for this suggestion and have now included a secondary analysis of previously-published snRNAseq data by Sarvari et al 2021 [PMID: 33378646], where we examined Tgm2

expression in multiple eWAT cell populations. Data have been included as Fig 1D- Fig 1E. Furthermore, we have sorted eWAT SVF cell populations from CD vs. HFD mice and performed qPCR analysis of Tgm2 expression within these cells. Data have been added as Fig 1I. Manuscript text has been revised, as follows:

Introduction:

[Lines: 62-64] “Transglutaminase 2 (TGM2) is a protein ubiquitously expressed in multiple cell types, including monocytes, M2 macrophages^{14, 15, 16, 17, 18, 19}, thymocytes²⁰, myoblasts²¹, endothelial cells²², and preadipocytes²³ among others.”

Methods:

[Lines: 231-232] “SVF was harvested from eWAT, as described³⁵ and utilized for: 1) staining of F4/80+TGM2 ATMs...”

[Lines: 245-257] “For staining of eWAT-derived SVF ATMs sorting: Panel1 (Used for protein lysate): Propidium Iodide (Thermo Fisher; Cat# P1304MP), CD45- Alexa Fluor 660 (Thermo Fisher; Cat#606-0451-82), MHC Class II – Brilliant Violet 650 (Biolegend; Cat#107641), F4/80-PE/Cy.7 (Biolegend; Cat#123113), CD11b-PerCP/Cyanine5.5 (Biolegend; Cat#101227), CD11c-Brilliant Violet 421 (Biolegend; Cat#117329). Cells were collected in PBS and further lysed in Nonidet P-40 lysis buffer with protease inhibitor cocktail, EDTA for 1h on ice before high-speed centrifugation; supernatant was collected as protein lysate. Panel 2 (Used for RNA isolation): Live/Dead- V450 (eBioscience; Cat#65-0863-14), CD45-AFF660 (Fisher scientific; Cat#606-0451-82), MHC Class II- BV510 (Biolegend; Cat#107641), F4/80-PE (Biolegend; Cat#111603), CD14-APCCy.7 (Biolegend; Cat#123317), TCRb-PECy.7 (Biolegend; Cat#109222), CD31-PerCPCy5.5 (Biolegend; Cat#160206), CD140a-BV605 (Biolegend; Cat#135916); cells were sorted directly into TRIzol reagent for subsequent RNA isolation”

[Lines: 223-228] “2.13 eWAT snRNAseq secondary analysis: We downloaded processed Seurat objects from Single Cell Portal accession ID SCP1179 generated by Sarvari et al 2021³⁴ containing data from 8355 immune cells (‘eWAT_Immune’). 309 out of 8355 cells (3.69%) expressing Tgm2 were labeled as Tgm2+. Cells with zero Tgm2 expression were labeled as Tgm2-. We then performed differential expression comparing Tgm2+ cells to Tgm2- cells using the Wilcoxon rank-sum test via the FindMarkers function of the Seurat R package v4.3.0.”

Results:

[Lines: 323-328] “To determine the source of AT TGM2 in HFD-induced obesity, a secondary analysis of snRNAseq eWAT published data by Sarvari et.al.³⁴, showed an increased abundance of Tgm2 expressing cells belonging to the immune cell fraction in HFD mice compared to CD group (**Fig. 1D**). Importantly, when narrowing down the analysis to focus on the immune cell populations, ATMs were the subset that showed a higher number of cells positive for Tgm2 in the HFD group compared to CD control (**Fig. 1E**).”

[Lines: 329-334] “Corroborative flow cytometric assessment in SVF from CD or HFD mice (gating strategy shown in **Supplementary Fig. S2B**) showed an increased abundance of F4/80+ ATMs (**Fig. 1G**), that co-expressed TGM2 (**Fig. 1H**) (as tested with a specific TGM2 antibody shown in **Supplementary Fig. S2C-S2E**). There was an increased number of TGM2+ATMs in the eWAT from HFD mice compared to CD controls.”

[Lines: 336-343] “Furthermore, to investigate if intracellular levels of Tgm2 differed in CD vs. HFD eWAT-derived cell populations, eWAT-derived SVF cells were sorted for preadipocytes, macrophages and monocytes (gating strategy in **Supplementary Fig. S2F**). No significant differences in intracellular Tgm2 expression by qPCR analysis were observed in CD vs. HFD preadipocytes or macrophages, while a significant increase in intracellular Tgm2 was shown in HFD eWAT-SVF sorted monocytes, compared to CD controls (**Fig. 1I**). This indicates that most of the increased TGM2 in the eWAT from HFD male mice emanates from an increased number of TGM2+ATMs and the presence of monocytes expressing high intracellular levels of Tgm2 compared to CD controls.”

Discussion:

[Lines: 578-585] *“Although our studies focused mostly on TGM2 expression in eWAT-derived SVF leukocytes, our secondary analysis of eWAT snRNAseq data showed Tgm2 expression in FAP, adipocyte, and endothelial cell populations from both CD and HFD eWAT, indicating that ATMs are not the sole contributor of TGM2 in AT during HFD conditions. However, it was the Tgm2+ATMs population that showed an obvious increase in cell numbers in the HFD group compared to CD control, suggesting that the increased macrophage population in eWAT from HFD serves as an extra source of TGM2 that significantly augments the overall expression of TGM2 at the tissue level.*

[Lines: 591-593] *“Further studies modulating TGM2 expression in other eWAT cell populations, or in the liver would greatly contribute to delineating the holistic role of TGM2 in diet-induced obesity metabolic dysfunction.”*

[Lines: 608-618] *“However, our data evaluating Tgm2 intracellular expression levels in monocytes also showed increased Tgm2 expression in HFD eWAT monocytes, compared to CD controls. Literature reports have previously shown the direct role of TGM2 in regulating monocyte adhesion and extravasation properties during high inflammatory settings¹⁶. Importantly, increased levels of TGM2 expression in monocytes leads to enhanced macrophage differentiation^{18, 50}, which would serve as a strong indication that TGM2+ATMs may result from an increased monocyte-derived infiltrating population that begins to have increasing Tgm2 expression along with macrophage differentiation/maturation/activation while residing in AT. A more detailed study ascertaining changes in the AT infiltrating monocyte population at different stages of the macrophage differentiation process would help test this hypothesis.”*

COMMENT #3) Furthermore, is the increase of TGM2 with obesity unique for the adipose tissue or does it also occur in other metabolic organs and tissues (liver, muscle)?

Response: This an excellent question. Although our original focus was on the epididymal white adipose tissue, we have now evaluated TGM2 protein expression in liver, kidney and pancreas from CD vs. HFD mice. We have included data as Supplemental Figure S2A, where we observed TGM2 to be increased in the liver from HFD mice compared to CD controls, but no change was seen in kidney or pancreas. We have updated manuscript text to address this observation, as follows:

Results:

[Lines” 311-315] *“To investigate how TGM2 body tissue expression is altered in these HFD mice, we evaluated TGM2 protein expression in metabolically relevant tissues. TGM2 protein expression was significantly increased in the liver, but not in the kidney or pancreas in HFD mice compared to CD controls (**Supplementary Fig. S2A**). Notably, HFD-treated mice had a two-fold increase in total AT transglutaminase activity.”*

Discussion:

[Lines: 585-593] *“Additionally, our data also showed increased expression of TGM2 in the liver from HFD male mice compared to CD controls, suggesting that TGM2 in liver tissue and/or liver macrophages could also potentially contribute to the modulation of systemic inflammation and tissue growth. Increased hepatic expression of TGM2 post-infection has been shown to be fundamental for development of liver fibrosis⁴⁸; thus, TGM2 could be a potential candidate responsible for regulating liver steatosis/fibrosis status in obesity-induced chronic inflammation. Further studies modulating TGM2 expression in other eWAT cell populations, or in the liver would greatly contribute to delineating the holistic role of TGM2 in diet-induced obesity metabolic dysfunction.”*

COMMENT #4) Transcriptomic analysis of TGM2+ and TGM2- ATMs could provide insight in the TGM2-dependent ATM plasticity.

Response: We concur with this notion and have now included such data as a secondary analysis of snRNAseq published data by Sarvari et al 2021 [PMID: 33378646] (Figure 1J). Manuscript text has been revised, as follows:

Methods:

[Lines: 223-228] “2.13 eWAT snRNAseq secondary analysis: We downloaded processed Seurat objects from Single Cell Portal accession ID SCP1179 generated by Sarvari et al 2021³⁴ containing data from 8355 immune cells (‘eWAT_Immune’). 309 out of 8355 cells (3.69%) expressing *Tgm2* were labeled as *Tgm2+*. Cells with zero *Tgm2* expression were labeled as *Tgm2-*. We then performed differential expression comparing *Tgm2+* cells to *Tgm2-* cells using the Wilcoxon rank-sum test via the FindMarkers function of the Seurat R package v4.3.0.”

Results:

[Lines: 344-349] “Lastly, to investigate the inflammatory profile of *Tgm2-* vs. *Tgm2+* eWAT ATMs in HFD male mice, we re-evaluated the snRNAseq data for CD and HFD mice published by Sarvari et.al.³⁴, focusing on the differential gene expression profile of the two populations. This data analysis found that cathepsin D (*Ctsd*), cathepsin L (*Ctsl*), and Matrix metalloproteinase-12 (*Mmp12*) were increased in the *Tgm2+* ATMs compared to the *Tgm2-* ATMs, while solute carrier family 9 member A9 (*Slc9a9*) was reduced (**Fig. 1J**).”

Discussion:

[Lines: 646-663] “Our further analysis investigating the transcriptional profile of *Tgm2+* ATMs identified *Ctsd*, *Ctsl*, and *Mmp12* genes to be significantly upregulated in the HFD *Tgm2+* ATMs when compared to *Tgm2-* subpopulation. Importantly, macrophage-derived *Ctsd* has recently been identified as a key suppressor of liver fibrosis via modulation of collagen remodeling and immune responses in vivo⁵⁵. Similarly, *Ctsl* expression was shown to be elevated in the lipid-associated macrophage (LAM) population⁵⁶, which is well-characterized by its anti-inflammatory effects in regulating phagocytosis and endocytosis leading to the idea that the LAM populations possesses a protective role in obesity-induced unhealthy AT^{57, 58}. In the case of *Mmp12*, an important regulator of extracellular matrix and wound healing responses that degrades basement membrane laminin⁵⁹, a previous ablation study showed its role in modulating endothelial cell dysfunction via increased extracellular matrix accumulation during tissue fibrosis stage. Conversely, *Slc9a9* gene expression was shown to be significantly downregulated in the *Tgm2+* ATMs subset, indicating the potential for this ATM subset to possess pro-inflammatory activities, at least in the context of bacterial killing, given *Slc9a9*’s role in modulating phagosome maturation via altering of the luminal PH⁶⁰. Altogether, *Tgm2* expression in ATMs seems to provide pro-resolving macrophage properties involved in increased accumulation of tissue extracellular matrix, which is commonly linked to increased tissue fibrosis status that results in insulin resistance and metabolic dysfunction⁶¹.”

COMMENT #5) Figure 1 should be in the supplement

Response: We have moved original Figure 1 as new Supplemental Figure S1 and manuscript text has been revised, accordingly.

Minor comments: The manuscript requires thorough editing, some examples:

Response: Thank you, we have heavily revised the manuscript to improve verbiage and clean text.

Line 26: ‘stromal vascular fraction (SVF) undergoing cell activation.’: unclear what is meant by ‘cell activation’.

Response: We meant to indicate that co-cultured cells were treated with anti-CD3/28 T cell activation beads. We have simplified this with the following revision:

[Line: 26-27] “... as well as in co-cultured eWAT stromal vascular fraction (SVF) cells.”

Line 37: ‘with inflammatory-driving immune cells’: ‘pro-inflammatory immune cells’.

Response: We have revised text, as follows:

[Lines: 42-43] “with pro-inflammatory immune cells”

Lines 186-7: ‘IL-10....CD45, TCRβ, CD4, 187 CD25, IL-10,’.

Response: Apologies for this revision error. We have fixed this methods section reflect all employed antibody panels, as follows:

Materials and Methods:

[Lines: 231-270]: “**2.14 Flow cytometry and AT Cells Sorting:** SVF was harvested from eWAT, as described³⁵ and utilized for: 1) staining of F4/80+TGM2 ATMs, 2) co-culture studies or 3) AT cell sorting. For staining approach of SVF, co-cultured or rTGM2-treated SVF, cells were harvested and incubated in Fc block Anti-Mouse CD16/CD32 (BD Biosciences; Cat#553141) 1:50 for 5 min on ice and washed 1x in FACS. Next, cells were incubated in primary conjugated antibodies diluted to 1:100 in FACS, unless otherwise specified for 20min at 4°C, as follows: MHC Class II-PerCp-eFluor710 (Thermo Fisher; Cat#46-5321-82), F4/80-PE Cy.7 (Biolegend; Cat#123113), CD206-Brilliant Violet 421 (Biolegend; Cat#141717), CD11c-Alexa Fluor 594 (Biolegend; Cat#117346), TCRβ-PE Cy.7 (Biolegend; Cat#109222), CD45- Alexa Fluor 660 (Thermo Fisher; Cat#606-0451-82), CD4-Alexa 488 (Biolegend; Cat#100425), CD25-PE Cy.5 (Biolegend; Cat#102010), IFNγ-PE/Dazzle (Biolegend; Cat#505846), IL-10- APC Cy.7 (Biolegend; Cat#5050335), IL-10- BV421 (Biolegend; Cat#505022), TCRβ- PercP Cy5.5 (Biolegend; Cat#109227), IFNγ-BV421 (Biolegend; Cat#505022).

For staining of eWAT-derived SVF ATMs sorting: Panel1 (Used for protein lysate): Propidium Iodide (Thermo Fisher; Cat# P1304MP), CD45- Alexa Fluor 660 (Thermo Fisher; Cat#606-0451-82), MHC Class II – Brilliant Violet 650 (Biolegend; Cat#107641), F4/80-PE/Cy.7 (Biolegend; Cat#123113), CD11b-PerCP/Cyanine5.5 (Biolegend; Cat#101227), CD11c-Brilliant Violet 421 (Biolegend; Cat#117329). Cells were collected in PBS and further lysed in Nonidet P-40 lysis buffer with protease inhibitor cocktail, EDTA for 1h on ice before high-speed centrifugation; supernatant was collected as protein lysate. Panel 2 (Used for RNA isolation): Live/Dead- V450 (eBioscience; Cat#65-0863-14), CD45-AFF660 (Fisher scientific; Cat#606-0451-82), MHC Class II- BV510 (Biolegend; Cat#107641), F4/80-PE (Biolegend; Cat#111603), CD14-APCCy.7 (Biolegend; Cat#123317), TCRb-PECy.7 (Biolegend; Cat#109222), CD31-PerCPCy5.5 (Biolegend; Cat#160206), CD140a-BV605 (Biolegend; Cat#135916); cells were sorted directly into TRIzol reagent for subsequent RNA isolation.

For staining of bone marrow macrophages: CD206-BV605 (Biolegend; Cat# C068C2), IL-10-APC Cy.7 (Biolegend; Cat#5050335), CD64-FITC (Thermo Fisher; Cat#MA5-46784), MHC Class II-Brilliant Violet 510 (Biolegend; Cat#107641), F4/80- Alexa Fluor 594 (Biolegend; Cat#123140). Cells were then washed in 100ul FACS 3x and re-suspended in 100uL 4%PFA for 30min on ice. Next, cells were permeabilized with 0.2% saponin and further resuspended in primary unconjugated TGM2 antibody (Thermo Fisher; Cat# MA5-12739) 1:500 at 2ug/mL or Mouse IgG1 kappa isotype control (eBioscience; Cat#14-4714-82) 1:500 at 2ug/mL overnight at 4°C. Cells were then washed in 0.2% saponin 3x and incubated for 30min on ice with secondary antibody targeting TGM2 or isotype control, Goat anti-Mouse, Alexa Fluor 488 (Thermo Fisher; Cat# A-21238) diluted to 1:5000 in 0.2% saponin. Cells were washed and re-suspended in FACS buffer for data acquisition. Flow cytometric data were analyzed using FlowJo software (v10.8.1) and represented as cells% and/or cells/g, as described. Mouse IgG1 kappa Isotype Control was used to determine TGM2 staining gating strategy.”

Lines 337-339: The designed studies and presented data in this paragraph, do not answer the question whether TGM2 is the only macrophage-derived key intermediary inducer of anti-inflammatory properties. The possible mechanisms mediating the effects of TGM2 could be discussed in the Discussion section based on reports in other cell types.

Response: We fully agree with reviewer. The goal of this experiment was to evaluate TGM2 as a potential soluble factor responsible for previously observed changes. We have also discussed TGM2 expression in other cell types in results and discussion sections, as shown in Major Comment#2. We have revised text to clarify purpose of in vitro recombinant TGM2 treatment, as follows:

[Lines: 468-469] “...to further investigate if TGM2 was a key soluble factor inducing anti-inflammatory properties ...”

Reviewer #2 (Remarks to the Author):

Concern #1: "...However, in some cases data should be improved by increasing the sample number to achieve more robust findings, for instance data shown in Figures 6H and S6H."

Response: We agree with this concern and have now included 2 additional biological replicates to the histology analysis in Figures 6H and S6H.

Concern #2: "In many cases, values are not normally distributed, for instance in Figure S6J, what statistical test was performed there?"

Response: Thank you for raising this important question. We evaluated normality on each data set via Shapiro-Wilk test, the Anderson-Darling test, Kolmogorov-Smirnov or D'Agostino & Pearson tests as required by the data. Unpaired Student t-test and One-Way ANOVA were used for normally distributed data. Mann-Whitney tests were employed for data not normally distributed. We have updated figure legends to specify test used in each figure data set.

Concern #3: The Cre mouse shown in Figure 6B does not seem to have the representative weight, which should be around 35 g (based on the graph in 6A) but rather a much bigger. Similarly, for the female mice, the Cre mouse in Figure S6B does not seem to have the weight of the average shown for Cre mice in S6A is around 25 g).

Response: Thank you for this important comment. We have replaced DEXA images in Figure 5B and S6B with an image from a more representative mouse for each group.

Concern #4: Y axis labelings in Figure 6C-F and S6C-F do not make any sense (Grams (g)).

Response: Apologies for this error – we fixed Y axis in Figure 6C and S6C-F to denote proper measuring condition and respective units.

Concern #5: The conclusion that Cre mice have less adipocytes is wrong, the Cre mice might actually have in total even more adipocytes than control mice, what the authors mean is that the number of adipocytes per ROI is less.

Response: We concur with reviewer and have revised the conclusion to focus attention on the hypertrophy effect showing fewer adipocytes per ROI.

[Line #1085-1087] "*Further histological assessment of eWAT stained for H&E (Fig. 6G) showed a significant increase in adipocyte size in the pLV.CD11b-Cre compared to pLV.Control male mice (Fig. 6H).*"

Concern #6: The nomenclature of genes and proteins must be corrected throughout the text and the figures.

Response: We believe we have now consistently written all genes and proteins in respective correct format throughout the text

Concern #7: Figure legends should state the sample numbers.

Response: We have included sample numbers for each data set in all figures.

Concern #8: The discussion must be shortened and focused on the most important points.

Response: Thank you – we have revised discussion and removed redundancy and sections non-relevant to the main focus of the study.

Concern #9: In the discussion the authors mention, that the TGM2 concentration used in in vitro experiments (1 µg/ml) may be similar to the endogenous concentration. To support this, the endogenous concentration in the adipose tissue must be determined.

Response: Thank you for pointing out this biased assumption based simply on increased overall TGM2 protein levels by western blot and/or qPCR in HFD adipose tissue compared to CD controls, which don't necessarily represent increased secretion within the adipose tissue. We have revised text by removing this assertion.

Concern #10: The authors also mention that no difference was observed in IL10+ ATMs between Cre and control mice potentially due to T cell exhaustion. Is T cell exhaustion setting in already at 6 weeks of HFD?

Response: Thank you for raising this important concern. We have come across a previous report demonstrating that although adipose tissue T cell exhaustion occurs at 12 and 18 weeks of HFD treatment via dysregulation of CD25 expression, adipose tissue T cells show inability to secrete IFN-γ or IL-2 upon stimulation already at 6 weeks of HFD [PMID: 33724954], demonstrating a potential start of T cell dysfunction. In line with this, adipose tissue T regs from mice placed on HFD for 4 weeks showed T cell-cycle arrest and increased cell death profile compared to CD, suggesting that adipose tissue T cell dysfunction process could start as early as 4 weeks of HFD exposure [PMID: 34256015].

Concern #11: Finally, as stressed out in the first round of revisions, the text must be carefully edited, below some examples, but authors should not limit their corrections to these:

line 294 'c'

line 349: S2F should be S3F

line 369: S3A should be S4A

line 730 'loss TGM2 loses'

Response: Thank you for identifying these mistakes – we have revised the manuscript and supplemental document file to correct mistakes in text.